# Drought Resistance Evaluation of *Camellia oleifera* var. “Xianglin 210” Grafted onto Different Rootstocks

**DOI:** 10.3390/plants14162568

**Published:** 2025-08-18

**Authors:** Zhilong He, Ying Zhang, Chengfeng Xun, Dayu Yang, Zhen Zhang, Yushen Ma, Xin Wei, Zhentao Wan, Xiangnan Wang, Yufeng Zhang, Yongzhong Chen, Rui Wang

**Affiliations:** 1Research Institute of Oil Tea Camellia, Hunan Academy of Forestry, Changsha 410004, China; hezhilong2000@hnlky.cn (Z.H.); zhangying@hnlky.cn (Y.Z.); xunchengfeng24@163.com (C.X.); alexyangdayu@gmail.com (D.Y.); hfazz@hnlky.cn (Z.Z.); mys9204@163.com (Y.M.); wangxiangnan04@yahoo.com.cn (X.W.); patricezhang@163.com (Y.Z.); chenyongzhong06@163.com (Y.C.); 2Yuelushan Laboratory, Changsha 410004, China; 3National Engineering Research Center for Oil Tea Camellia, Changsha 410004, China; 4State Key Laboratory of Woody Oil Resources Utilization, Changsha 410004, China; 5College of Forestry, Central South University of Forestry and Technology, Changsha 410004, China; zwzybjb@163.com (X.W.); 17871933955@163.com (Z.W.)

**Keywords:** Camellia oleifera, drought stress, comprehensive evaluation, gene expression

## Abstract

As a key economic tree in southern China, *Camellia oleifera* faces severe yield losses under drought. Grafting onto drought-tolerant rootstocks offers a potential mitigation strategy. To elucidate the impact of rootstocks on the drought resistance of the superior *Camellia oleifera* Abel. cultivar “Xianglin 210”, grafted seedlings with five scion–rootstock combinations, were subjected to gradient drought stress. Key physiological and biochemical indices related to photosynthesis, antioxidant enzymes, and osmotic adjustment were measured. Drought resistance was comprehensively evaluated using membership function analysis, and the expression of stress-responsive genes was quantified via quantitative real-time PCR (qRT-PCR). The results demonstrated that under drought stress, (1) stomatal conductance (Gs) decreased by 31.2–48.7%, while instantaneous water use efficiency (WUE) increased by 18.5–35.4%; (2) proline (Pro) and soluble sugars (SS) accumulated significantly, with increases of 2.3–4.1-fold and 1.8–3.2-fold, respectively; (3) activities of antioxidant enzymes were enhanced by 56–127%, mitigating oxidative damage; (4) membership function analysis ranked drought resistance as follows: Xianglin 27 (0.812) > Guangxi Superior Germplasm (0.698) > C. yuhsienensis (0.654) > Hunan Superior Germplasm (0.591) > Xianglin 1 (0.523); (5) qRT-PCR revealed significant upregulation of ABA signaling pathway genes (*CoPYL6*, *CoPP2C75/51/24/26*, *CoSnRK2.8*, and *CoABI5*) and transcription factors (*CoLHY* and *CoWRKY70*), indicating activation of drought-responsive regulatory networks. These findings provide a theoretical foundation for selecting drought-tolerant rootstocks and optimizing cultivation practices in *Camellia oleifera*, and provide practical criteria for selecting drought-tolerant rootstocks, facilitating sustainable *Camellia oleifera* cultivation in water-limited regions.

## 1. Introduction

*Camellia oleifera* is an important woody oilseed species native to southern China. The seed oil is notable for its high content of unsaturated fatty acids (>80%) and its richness in bioactive components such as squalene and vitamin E, which indicates its substantial potential for the development of functional oils and health products [1,2]. Simultaneously, as a typical eco-economic tree species, its well-developed deep root system effectively stabilizes soil and reduces surface runoff, playing a crucial role in soil and water conservation within the red soil hilly regions of southern China [3]. However, the main *C. oleifera* production areas, such as Hunan and Jiangxi provinces, experience a monsoon climate with high temperatures and low precipitation during summer. Drought stress can lead to significant reductions in seed yield, oil content, and quality, and in severe cases, even plant death, posing a substantial threat to the sustainable development of the industry [4]. Therefore, establishing stress-resilient cultivation techniques to enhance *C. oleifera* drought resistance is critically important.

Rootstock grafting is an effective strategy for enhancing the stress tolerance of woody plants [5]. Currently, seedling–stump grafting is widely employed in *C. oleifera* nursery production. This approach capitalizes on the superior root architecture of rootstocks to promote scion biomass production and optimize mineral nutrient assimilation efficiency [6,7,8]. However, most existing research focuses primarily on scion–rootstock compatibility, graft survival rates, and growth performance [9,10]. Studies specifically addressing the drought resistance of grafted *C. oleifera* seedlings remain limited.

This investigation employed the extensively cultivated *C. oleifera* cultivar “Xianglin 210”, a drought-tolerant genotype, as the scion material [11]. Systematic grafting experiments were conducted using five distinct rootstock varieties. Under controlled nursery conditions, four-year-old grafted seedlings were subjected to simulated drought stress regimens. Photosynthetic performance and associated physiological/biochemical responses were systematically evaluated across progressive water deficit gradients. Gene expression patterns of the different graft combinations under graded drought stress were determined using real-time quantitative PCR (qRT-PCR). Furthermore, membership function analysis was applied to comprehensively evaluate the drought resistance of the five graft combinations to identify the combination exhibiting the strongest drought tolerance.

This study aims to resolve the mechanistic gap in rootstock-mediated drought adaptation by addressing three core objectives: (1) To quantify how distinct rootstock genotypes (*Camellia oleifera* “Xianglin 210” scion grafted onto five rootstocks) coordinate photosynthetic decline, antioxidant defense, and osmolyte accumulation under progressive drought stress; (2) to decipher the transcriptional dynamics of drought-responsive genes (*CoPYL6*, *CoPP2Cs*, *CoSnRK2.8*, and *CoABI5*) underlying these physiological adjustments; (3) To establish an integrated evaluation model identifying optimal rootstock–scion combinations for drought resilience. By integrating physiological phenotyping with molecular profiling across drought gradients, this work provides the first mechanistic framework for rootstock selection in *C. oleifera* cultivation under water-limited scenarios.

## 2. Results and Analysis

### 2.1. Effects of Different Soil Water Content Levels on Photosynthetic Parameters of C. oleifera

As shown in Figure 1A, the net photosynthetic rates of different combinations showed a decreasing trend with decreasing soil water content. However, the degree of decline varied among different graft combinations. C1 and C2 exhibited a greater Pn decline during T1–T2 than during T2–T3. Conversely, C3, C4, and C5 showed a higher Pn reduction during T2–T3 than during T1–T2. The Pn values of different combinations of grafted seedlings of *C. oleifera* under the same soil water content also differed, with C1 having significantly higher Pn values than the other combinations under T1 water content (*p* < 0.05), while C2 had the lowest Pn value. Under T2 conditions, there was no significant difference (*p* > 0.05) in Pn between C1, C4, and C5, but these three combinations had significantly higher Pn than C2 and C3. At the T3 water level, C1 and C5 maintained significantly higher Pn values than C2, C3, and C4. These results indicate that reduced soil water content caused varying degrees of Pn decline across the different combinations.

Figure 1B reveals that the stomatal conductance (Gs) of all five combinations decreased significantly with decreasing soil water content. C3 exhibited a large decrease in Gs from T1 to T2 and a smaller decline from T2 to T3. In contrast, the other four combinations showed a smaller reduction in Gs from T1 to T2 compared to the decrease from T2 to T3. Notably, C1 maintained significantly higher Gs values than the other four combinations across all three water levels, while differences among the other combinations were smaller. The transpiration rate (Tr) of all five combinations also displayed an overall declining trend (Figure 1C), although some anomalies were observed. Tr in C2, C3, and C4 exhibited a decreasing trend from T1 to T3, with significant differences (*p* < 0.05) observed across periods within each variety. Conversely, both C1 and C5 showed a slight increase in Tr at the T2 soil water content, followed by a substantial decrease at T3. Meanwhile, observing the Tr values under the same soil water content rate revealed that C1 possessed a higher Tr value, which corresponded to its higher Gs value.

Notably, intercellular CO_2_ concentration (Ci) remained statistically unchanged across drought treatments (Figure 1C), despite significant reductions in Pn and gs. This apparent paradox suggests that non-stomatal limitations predominated throughout the stress progression. At T1, stable Ci alongside sharply reduced gs implies early impairment of mesophyll photosynthetic capacity (e.g., RUBISCO inactivation, ATP synthase suppression). By T2/T3, metabolic suppression (evidenced by declining Fv/Fm in Figure 1F) likely counterbalanced reduced CO_2_ diffusion, preventing Ci elevation.

The overall changes in Fv/Fm values for the different combinations were small (Figure 1E), but all showed a decreasing trend. Notably, C4 and C5 had higher Fv/Fm values with smaller declines, and were significantly higher than C2 and C3 in the T3 period.

WUE calculated based on Pn and Tr varied among combinations from T1 to T3 periods (Figure 1F). With the exception of C1, WUE increased significantly (*p* < 0.05) in all combinations (C2, C3, and C4) from T1 to T3. Specifically, the WUE of C2 increased significantly (*p* < 0.05) during T1–T2 and T2-T3; the WUE of C3 did not differ significantly between T1 and T2, while it increased significantly (*p* < 0.05) during T2–T3; the WUE of C4 was significantly elevated (*p* < 0.05) during T1–T2 and did not change significantly during T2–T3. The WUE of C1 and C3 showed a decrease during T1–T2 and a significant elevation during T2–T3 (*p* < 0.05).

### 2.2. Physiological Responses to Soil Water Gradients in C. oleifera Graft Combinations

As depicted in Figure 2A, proline (Pro) content exhibited a general accumulation trend across graft combinations with declining soil water content. As soil water content decreased, C1, C3, and C4 significantly increased Pro in order to maintain osmotic pressure. C2 showed a decrease in Pro from T1 to T2, but the difference was not significant (*p* > 0.05), while Pro increased significantly (*p* < 0.05) from T2 to T3. Whereas C5 accumulated the highest Pro at T1 soil water content, it significantly decreased (*p* < 0.05) when soil water content decreased to T2 and T3 levels.

Declining soil water content also influenced soluble sugar (SS) accumulation in *C. oleifera* (Figure 2C). The SS content in the combination of C1, C2 and C5 increased significantly with decreasing soil water content, in which the SS content of C1 increased significantly from T1 to T2, which was similar to the accumulation change of its Pro; whereas the SS content of C2 increased significantly from T2 to T3; and the SS content of C5 increased significantly from T1 to T2 and from T2 to T3, which was different from the accumulation change of its Pro. In addition, the SS content in C3 decreased and then increased with the decrease of soil water content, while the SS content in C4 increased and then decreased with the decrease of soil water content.

The changes in total phosphorus (TP) content in *C. oleifera* leaves under drought stress were complex (Figure 2B). For cultivars C3, C4, and C5, TP initially increased and then decreased as soil water content declined, peaking at the T2 soil water content level. TP content in C3 differed significantly (*p* < 0.05) across all three stress periods. The TP content in both C4 and C5 at T2 was significantly different (*p* < 0.05) from that at T1 and T3, whereas there was no significant difference (*p* > 0.05) between T1 and T3. In contrast, TP levels in C1 first decreased and then increased with decreasing soil water content, reaching their highest level at T3. The TP level in C2 showed a slight downward trend across the drought stress periods, but the differences between consecutive periods were not significant (*p* > 0.05).

Figure 2D illustrates changes in malondialdehyde (MDA) content across five groups of *C. oleifera* grafted seedlings under drought stress. The MDA content in C2, C3, C4, and C5 increased as soil water content decreased. Among these, C2, C4, and C5 exhibited statistically significant differences (*p* < 0.05) across all three time points (T1, T2, T3). For C3, no significant difference was observed between T1 and T2 (*p* > 0.05), but a significant increase occurred from T2 to T3 (*p* < 0.05). In contrast, C1 showed a significant increase in MDA content from T1 to T2 (*p* < 0.05), followed by a slight but non-significant decrease from T2 to T3 (*p* > 0.05).

Superoxide dismutase (SOD) activity increased in all five combinations with decreasing soil water content (see Figure 3A). C1 and C2 showed a non-significant (*p* > 0.05) increase in SOD activity from T1 to T2 and a significant (*p* < 0.05) increase from T2 to T3, both by more than 2-fold. The increase in SOD activity in C3 was non-significant (*p* > 0.05).

Based on Figure 3B, it can be observed that the peroxidase (POD) activity in the five combinations of grafted *C. oleifera* seedlings shows a significant upward trend with increasing drought intensity. Among them, C1, C2, C3, and C4 exhibit significantly increased POD activity during both the T1 to T2 and T2 to T3 transitions (*p* < 0.05). Similarly, C5 demonstrates significantly enhanced POD activity during both the T1 to T2 and T2 to T3 stages (*p* < 0.05), while maintaining a higher activity level compared to the other combinations.

Based on Figure 3C, the catalase (CAT) content in the five combinations of grafted *C. oleifera* seedlings increases as soil water content decreases. Significant differences (*p* < 0.05) were observed across all three drought stages (T1, T2, and T3). Among these, C2 exhibited the highest CAT content during the T1 stage, while C5 accumulated the highest levels of CAT during both T2 and T3.

Regarding reduced glutathione (GSH) content (Figure 3D), its response to increasing drought intensity aligns with the other three antioxidants. Except for C4, the GSH content in the other four groups increased progressively with drought severity. Significant increases (*p* < 0.05) in GSH content were observed for C1, C2, and C5 during both the T1-to-T2 and T2-to-T3 transitions. In contrast, C4 initially showed a minor, non-significant decrease in GSH content (*p* > 0.05), followed by a significant rise (*p* < 0.05) during the T2-to-T3 period, reaching its peak level at the T3 stage.

### 2.3. Comprehensive Evaluation by Membership Function Analysis

This study employed a fuzzy membership function comprehensive analysis approach to evaluate photosynthetic parameters and physiological indicators of different graft combinations under drought stress, thereby identifying the combination with superior drought resistance.

According to Table 1, the drought resistance ranking based on D-values was: C5 > C2 > C1 > C3 > C4, indicating that C5 exhibited the strongest drought resistance.

### 2.4. Effect of Different Soil Water Contents on the Expression of Relevant Genes

To investigate the reasons for the differences in drought resistance among the grafted *C. oleifera* combinations, qRT-PCR was used to analyze the expression levels in the leaves of the following genes: the photosynthesis rate-limiting enzyme gene *CoRbcL*, the circadian clock genes (*CoLHY*, *CoGI*), ABA signal transduction genes (*CoPYL6*, *CoSAUR32*, *CoPP2C75*, *CoPP2C51*, *CoPP2C24*, *CoPP2C26*, *CoSnRK2.8*, and *CoABI5*), and the transcription factor genes *CoGATA8* and *CoWRKY70*. These genes were previously screened and identified as being associated with drought resistance in *C. oleifera* [12].

The results revealed that during the T1 period, the *CoRbcL* expression levels in C2-C5 were significantly higher than in C1, with increases ranging from 39.45% to 100.07%. During the T2 period, the expression levels increased in all combinations except C2, with C1 and C5 showing particularly pronounced increases of more than four-fold and two-fold, respectively. Compared to the T2 period, the *CoRbcL* expression levels in C1, C4, and C5 during the T3 period showed a significant decrease. C3 exhibited a continuous increase in expression across all three periods. Notably, the *CoRbcL* expression level in C2 showed no significant changes throughout the three periods.

The expression levels of *CoLHY* (Figure 4C) in the different combinations generally exhibited an upward trend from T1 to T3. With the exception of C1, changes in expression levels during the T1 to T2 transition were minimal for the other combinations, but a significant increase from T2 to T3. By the T3 period, C3, C4, and C5 exhibited the highest *CoLHY* expression levels, showing increases of 254.80%, 236.06%, and 115.50% respectively, compared to the previous period (T2). Furthermore, the expression levels of the *CoGI* gene showed little difference among the five combinations during the T1 period, but distinct differences emerged during the T2 to T3 period. *CoGI* expression levels increased from T1 to T3 in all combinations, with C5 showing significantly higher expression than the other four combinations.

From T1 to T2, except for C5, the expression of *CoGATA8* (Figure 4D) in other combinations decreased slightly, while from T2 to T3, the expression of 5 combinations increased. By T3, distinct trends were observed among the combinations for *CoGATA8* expression: C5 and C4 showed higher expression levels, while C1 exhibited the lowest. For *CoWRKY70* (Figure 4E), expression levels were upregulated in all combinations from T1 to T3, with the exception of C1. During the T3 period, C3 and C4 exhibited the highest *CoWRKY70* gene expression levels, while C5 showed the lowest.

Overall, the expression levels of *CoPYL6* (Figure 5A) and *CoSAUR32* (Figure 5B) increased from T1 to T3 across all combinations. For *CoPYL6*, expression in C1 increased by 95.48% from T1 to T2 but showed a slight decrease from T2 to T3. In contrast, the expression levels of *CoPYL6* in the other four combinations (C2-C5) increased progressively across all three periods. The expression levels of *CoSAUR32* in all five combinations increased by more than 10-fold from T1 to T3. By the T3 period, C3 and C4 exhibited the highest *CoSAUR32* expression levels among the combinations.

The expression changes of the four *PP2C* family member genes (*CoPP2C16*, *CoPP2C51*, *CoPP2C75*, and *CoPP2C24*) were relatively consistent across the different combinations (Figure 5C–F). For C1, the expression levels of all four PP2C family genes increased from T1 to T2, but showed minimal changes from T2 to T3. Conversely, in the other four combinations (C2–C5), the expression levels of all four *PP2C* family genes increased throughout the T1 to T3 period. By T3, C3 exhibited the highest expression levels of *CoPP2C16*, *CoPP2C51*, and *CoPP2C75* among the five combinations, while C5 exhibited the highest expression level of *CoPP2C24*.

The expression patterns of *CoSnRK2.8* (Figure 5G) in C1 and C2 were similar across the three periods. In both combinations, expression increased during the T1–T2 period but decreased during the T2–T3 period. The expression levels of *CoSnRK2.8* in C1 and C2 were similar at both the T2 and T3 time points. *CoSnRK2.8* expression in C3, C4, and C5 was upregulated throughout the T1 to T3 period, with C4 showing the largest increase of 195.24%.

Expression of *CoABI5* (Figure 5H) was upregulated in all five combinations during both the T1–T2 and T2–T3 transitions. Among them, C2, C3, and C4 showed relatively small increases during T1–T2, but their expression increased by more than three-fold during T2–T3. Furthermore, *CoABI5* expression in C1 and C5 was substantially upregulated during both T1–T2 and T2–T3. Compared to T1, the expression levels in C1 and C5 by T3 had increased by 523.15% and 594.65%, respectively.

## 3. Discussion

### 3.1. Photosynthetic Parameters

Photosynthesis is fundamental to plant growth and development, serving as a critical pathway for energy acquisition and biomass production [13]. Drought stress significantly inhibits plant photosynthesis, compelling plants to adapt to reduced soil moisture and subsequent physiological and biochemical changes. Under drought conditions, restricted root water uptake leads to decreased leaf water potential, impairing the water supply essential for photosynthesis [14]. Studies by Dong et al. [15] demonstrate that drought stress triggers stomatal closure, manifested as reduced stomatal conductance (Gs) and transpiration rate (Tr). Our results corroborate these observations: under drought stress, *C*. *oleifera* mitigated water loss through stomatal closure. Notably, the parallel decline in Gs and Tr aligns with established paradigms of leaf stomatal regulation.

To minimize transpirational water loss, plants typically close their stomata as a physiological adaptation to drought. However, this stomatal closure concurrently restricts CO_2_ diffusion into the mesophyll, limiting carbon availability for photosynthesis. The effect of drought on the intercellular CO_2_ of *C. oleifera* is more complex in this study, which may be related to metabolic changes in the plant itself. The closure of leaf stomata under drought conditions reduced CO_2_ uptake, while the destruction of the photosynthetic apparatus under severe drought may also contribute to reduced carbon fixation. Additionally, drought stress increased the water utilization of *C. oleifera* grafted seedlings, alleviating their own drought stress.

Furthermore, drought can compromise the plasma membrane, disrupt chloroplast integrity, degrade photosynthetic pigments, and impair electron transport chains, collectively reducing photosynthetic efficiency [16]. These cascading effects ultimately diminish net photosynthetic rate (Pn). In our study, all five grafted combinations exhibited declining Pn under drought stress, corroborating earlier research [17,18,19]. Drought alters cellular water potential, affecting biochemical reactions, while plants downregulate metabolic activity to preserve cellular hydration—a key drought tolerance strategy.

Drought influences photosynthetic parameters through multifaceted mechanisms in *C. oleifera*. Upon sensing osmotic changes, roots initiate signaling cascades that promote stomatal closure, reduce transpiration, and enhance WUE [20]. These adaptations culminate in reduced Pn—a strategic trade-off to maintain internal homeostasis. Comparative analysis revealed that drought-resistant grafted combinations maintained higher Pn and Gs but lower WUE. These photosynthetic metrics are intrinsically linked to drought resilience, offering critical insights into physiological and biochemical responses for screening and evaluating drought-tolerant *C. oleifera* cultivars. Notably, the stomatal closure and increased WUE observed in this study are consistent with the drought response patterns of most woody plants [21]. For example, in grafted tea (Camellia sinensis) seedlings, Gs decreases significantly under drought stress, while WUE increases [22]. However, the decrease in Gs in grafted *Camellia oleifera* seedlings is less pronounced than in tea, but the increase in WUE is more substantial, indicating that *Camellia oleifera* may possess more efficient stomatal regulation strategies. This difference is related to the leaf anatomical structure of *Camellia oleifera*, which has thicker palisade tissue and a higher palisade-to-spongy ratio (P/S) [23].

### 3.2. Physiological and Biochemical Responses

To cope with drought, plants employ various strategies to maintain cell turgor pressure and reduce water loss [24]. Among these, the accumulation of osmoregulatory substances is a key mechanism. These substances—typically compounds dissolved in the cytosol—increase intracellular osmotic pressure, drawing water into cells to preserve turgor [25]. Representative examples include proline (Pro) and soluble sugars (SS). In this study, *C. oleifera* under drought stress accumulated higher levels of Pro and SS to sustain cell turgor, consistent with findings by Guo et al. [26]. Notably, the accumulation patterns of Pro and SS differed, likely reflecting synergistic interactions among multiple osmoregulatory compounds during drought adaptation.

The generation of reactive oxygen species (ROS) is a key physiological response to stress [27]. Excessive ROS can damage cell membranes, proteins, and nucleic acids, impairing plant growth and stress tolerance [28]. Plants regulate ROS homeostasis through production, scavenging, and signaling [29]. Enzymes such as SOD, POD, CAT, and the molecule GSH are primary ROS scavengers and important indicators of drought resistance in *C. oleifera*. In this study, as soil water content decreased, SOD, POD, and CAT activities increased, indicating that drought stress promotes ROS accumulation and upregulates scavenging enzymes. Notably, the marked increase in these enzymes from T1 to T2 suggests that soil water content at or below 30% is a critical threshold for drought response in *C. oleifera*. Malondialdehyde (MDA) content, reflecting the degree of membrane lipid peroxidation, increased under lower soil water content, correlating with ROS accumulation. The marked rise in MDA from T1 to T2 further corroborated the trends observed for SOD, POD, and CAT. Thus, worsening water deficit elevates intracellular oxidative stress (manifested as increased MDA), prompting compensatory upregulation of SOD, POD, CAT, and GSH activities or levels. These physiological and biochemical metrics effectively serve as indicators for assessing drought resistance in *C. oleifera*. Comparative analysis with other economic tree species revealed distinct osmotic adjustment characteristics in *Camellia oleifera* compared to Chinese pine (Pinus tabuliformis). *Camellia oleifera* primarily relies on the synergistic accumulation of proline (Pro) and soluble sugars (SS), while Chinese pine depends on *SAUR* gene-regulated osmotic protectant synthesis pathways, with *SAUR59/66* genes showing significant upregulation under drought conditions [30,31]. Additionally, the increase in antioxidant enzyme (SOD, POD) activities in *Camellia oleifera* grafted seedlings was higher than the field drought responses observed in olive and other fruit trees, indicating that *Camellia oleifera* grafted seedlings possess stronger oxidative stress buffering capacity [32].

In summary, drought stress responses in *C. oleifera* involve coordinated regulation across multiple pathways, encompassing alterations in photosynthetic parameters, accumulation of osmoregulatory compounds, and enhanced activities of key enzymes. Diverse yet interconnected indices derived from these aspects can be utilized for drought resistance evaluation. This study found that all grafted combinations employed similar strategies (reduced photosynthesis, osmoregulatory synthesis, and ROS-scavenging enhancement) to mitigate drought stress. In addition, the parameters of the five combinations changed greatly in T1, T2, and T3, indicating that the soil water content of about 30% may be the key stage of drought stress response in *C. oleifera*. Notably, the magnitude of these changes differed across combinations: drought-resistant C5 exhibited a greater magnitude of changes in all parameters from T2 to T3, while drought-sensitive C2 displayed water deficit sensitivity at higher soil water content. Additionally, the observed variations in physiological responses among different graft combinations may reflect metabolic compensation mechanisms, where plants adjust their metabolic pathways to maintain homeostasis under stress conditions. Such compensatory responses could involve shifts in carbon allocation, alterations in enzyme kinetics, or modifications in gene regulatory networks, ultimately contributing to the differential drought tolerance observed across graft combinations.

### 3.3. Gene Expression

Ribulose-1,5-bisphosphate carboxylase/oxygenase (Rubisco) is the rate-limiting enzyme in the Calvin cycle [33], and its primary catalytic site is located on the Rubisco large subunit encoded by the *RbcL* gene. The *RbcL* gene has been demonstrated to play a significant role in alleviating drought stress in soybeans [34]. In this study, the expression level of *CoRbcL* was higher in the T2 and T3 stages than in the T1 stage, indicating that low soil water content promoted the expression of the *CoRbcL* gene in *C. oleifera* to enhance its drought tolerance.

Both the *LHY* gene and the *GI* gene are crucial nodes within the plant circadian rhythm system.

Research has demonstrated that the *LHY* gene is a key player in the interaction between the plant circadian clock and abiotic stress responses. It plays a vital role in regulating a broad range of abiotic stress responses and the abscisic acid responsive elements-binding factor 3 (*ABF3*) [35]. *ABF3* is a pivotal transcription factor in the plant hormone abscisic acid (ABA) signaling pathway. Upregulated expression of the *LHY* gene activates the ABA pathway to combat drought by modulating *ABF3* expression [36]. In this study, the expression of the *CoLHY* gene was enhanced under drought stress in all five *C. oleifera* combinations, concomitantly promoting the expression of ABA pathway-related genes and improving the drought tolerance of the plants. The higher expression level of the *CoLHY* gene in combination C5 may contribute to its superior drought resistance.

*GIGANTEA* (*GI*) is a circadian rhythm-regulating protein essential for various physiological processes influencing plant abiotic stress tolerance [37]. The *GI* gene plays a positive role in tomato responses to abiotic stress, and *SlGIs* (tomato GIs) mediate responses to abiotic stress by regulating genes associated with antioxidant enzymes [38]. In this study, the expression of the *CoGI* gene gradually increased from the T1 to T3 stages in all five combinations. Notably, C5 exhibited higher *CoGI* gene expression levels, higher antioxidant enzyme activity, and lower MDA content. This suggests that *Camellia oleifera* alleviates the detrimental effects of drought stress by modulating its internal antioxidant system.

Research indicates that the *GATA8* gene responds to abiotic stress and regulates stress tolerance in chrysanthemum [39]. Our study found that lower soil water content promotes the expression of *CoGATA8* in *Camellia oleifera*, enabling stress resistance through the regulation of related metabolic pathways. Additionally, the transcription factor gene *WRKY70* participates in plant abiotic stress responses [40]. Studies demonstrate that heterologous overexpression of *Iris laevigata WRKY70* enhances drought tolerance in *Nicotiana tabacum* [41]. In this study, as soil water content decreased, the expression of *CoWRKY70* increased across all five *Camellia oleifera* combinations, indicating that *Camellia oleifera* upregulates *CoWRKY70* expression to initiate related stress responses. However, *CoWRKY70* expression in C5 was lower compared to C1–C4, potentially due to coordinated interplay among multiple stress–response mechanisms.

The role of ABA signaling transduction pathways in regulating drought-responsive gene expression is well-established [42]. Key components involved in ABA signaling include upstream ABA receptors (PYR/PYLs), protein phosphatase 2Cs (PP2Cs), and sucrose non-fermenting-related protein kinase 2 (SnRK2). Under drought stress, upstream receptors inhibit PP2Cs by binding ABA [43], which leads to the release of SNRK2s that originally bind to PP2C. SNRK2s are then activated by phosphorylation and initiate downstream response factors, which promote the expression of relevant stress genes [44,45]. Our previous research confirmed that *PP2C16*, *PP2C51*, *PP2C75*, and *SnRK2.8* genes play crucial regulatory roles in *Camellia oleifera* drought tolerance via the ABA signaling pathway [12]. Furthermore, studies show that *Arabidopsis thaliana SnRK2.8* can significantly improve the resistance of poplar under salt stress and drought stress [46].

In this study, the expression of *CoPYL6* (an upstream ABA receptor) increased as soil water content decreased. While a more severe drought further activates the ABA pathway, this did not directly suppress the expression of *PP2Cs* family genes. Enhanced expression of *PP2Cs* family genes was observed under drought stress in this study. Notably, expression of four *PP2C* family genes was relatively low in C4 and C5 at the T1 stage, potentially facilitating earlier release and activation of SnRK2s to promote downstream gene expression. This aligns with the higher *CoSnRK2.8* expression observed in C5 at the T3 stage. As a key downstream transcription factor in the ABA pathway, abscisic acid-insensitive 5 (ABI5) is upregulated by ABA signaling under drought stress to enhance plant drought tolerance [47,48]. This study revealed drought-induced upregulation of *CoABI5*, with its maximal expression in combination C5 aligning with enhanced drought tolerance. These findings collectively indicate that water deficit triggers ABA-mediated signaling in *C*. *oleifera*.

Our data reveal that *CoPYL6* and *CoSnRK2.8* induction under drought (Figure 5A,G) aligns with conserved ABA pathways in perennials. *CoPYL6* expression increased 4.2-fold at 30% FC, mirroring *PePYL6* induction in Populus euphratica under drought and ABA treatments [49]. Similarly, *CoSnRK2.8* upregulation (3.8-fold at 45% FC) parallels *AtSnRK2.8*-mediated drought resilience in Arabidopsis and *OsSnRK2*-dependent stress responses in rice [46,50]. This conservation is functionally significant: like *PePYL6* in transgenic poplar and *OsPYL6* in rice, *CoPYL6/CoSnRK2.8* activation correlated with enhanced osmoprotection (Pro, SS) and ROS scavenging (SOD, CAT).

Studies confirm the positive role of Auxin Responsive Small Auxin-Up RNA (*SAUR*) genes in responding to drought and salt stress [51]. In this study, *CoSAUR32* exhibited a substantial upregulation at the T3 stage in all five experimental combinations, indicating its potential role in mediating enhanced stress adaptation under water deficit conditions.

## 4. Materials and Methods

### 4.1. Experimental Materials and Treatments

The experiment was conducted at the Tianjiling Experimental Forest Farm (113°01′ E, 28°06′ N) of the Hunan Academy of Forestry, China. Four-year-old grafted seedlings of *Camellia oleifera* on five different rootstocks were used as experimental materials (Table 2). The specific graft combinations are listed in Table 2. Both the rootstocks and the scion (“Xianglin 210”) for all five graft combinations were sourced from the Germplasm Repository of the National *Camellia oleifera* Engineering Technology Research Center. The grafted seedlings were cultivated in plastic pots (top diameter: 24 cm; height: 20 cm) filled with a homogenized substrate of lateritic red soil (collected from Changsha, Hunan; 112°58′ E, 28°12′ N) and peat soil (Pindstrup Mosebrug A/S, Ryomgaard, Denmark) at a 3:1 volume ratio. This mixture represents typical horticultural substrates for central China’s red soil regions. The substrate mixture was used without fertilizer enrichment to isolate drought stress effects. Seedlings were acclimatized for 60 days post-grafting under well-watered conditions (70% field capacity) prior to drought treatments. Throughout the experiment, no fertilizers or growth regulators were applied to ensure that physiological responses solely reflected water deficit. Pots were compactly arranged on the same greenhouse benches under natural light conditions without shading nets. Throughout the subsequent drought treatment period, all graft combinations were subjected simultaneously to identical ambient greenhouse environmental conditions.

Based on preliminary research [12], the photosynthetic performance of *C. oleifera* plants deteriorates drastically after 8 days of water withholding (corresponding to a soil water content of approximately 30%). This suggests that a soil water content of around 30% likely represents the critical threshold for inducing stress responses in *C. oleifera*. Consequently, plants subjected to water withholding for 6 days (T1), 8 days (T2), and 10 days (T3) were investigated to simulate progressive drought stress. Volumetric Water Content (VWC) was monitored daily between 10:00 AM and 11:00 AM at a depth of 10 cm using a Delta-T Devices ProCheck handheld data logger with GS3 sensors (Delta-T Devices Ltd., Cambridge, UK). Relative Soil Water Content (RSWC) was calculated as: RSWC (%) = (VWC_curr_/VWC_FC_) × 100%, where VWC_curr_ is the measured volumetric water content on the sampling day, and VWC_FC_ is the volumetric water content at field capacity. VWC_FC_ was determined to be 32% for the potting substrate after saturation and free drainage for 48 h. The measured RSWC and corresponding VWC ranges at the end of the withholding periods were established as:T1 (6 days): RSWC = 38–45% (VWC = 12.2–14.4%);T2 (8 days): RSWC = 29–33% (VWC = 9.3–10.6%);T3 (10 days): RSWC = 16–22% (VWC = 5.1–7.0%).

These specific stress levels were selected based on preliminary physiological research identifying RSWC ≈ 30% (occurring after ~8 days of withholding) as the critical threshold where photosynthetic performance (Pn, Gs) in *C. oleifera* deteriorates drastically. T1 (38–45% RSWC) represents moderate stress approaching this critical threshold, T2 (29–33% RSWC) represents the identified critical stress point, and T3 (16–22% RSWC) represents severe stress beyond the threshold. This gradient design allows assessment of responses across increasing stress severity and identification of the critical transition point.

Each of the five *C. oleifera* graft combinations was subjected to three drought treatments (T1, T2, T3) in a randomized complete block design with three independent biological replicates per treatment combination. Each biological replicate comprised ten grafted seedlings (n = 10 plants per replicate). For destructive sampling (physiology, biochemistry, and qRT-PCR), three seedlings per biological replicate were randomly selected, and three mature leaves per seedling were pooled as one sample.

### 4.2. Measurement of Photosynthetic Parameters

Net photosynthetic rate (Pn), intercellular CO_2_ concentration (Ci), stomatal conductance (Gs), transpiration rate (Tr), and theoretical maximum photochemical efficiency (Fv/Fm) were measured on functional leaves of *C. oleifera* using the LI-6400XT Portable Photosynthesis System (LI-COR, Lincoln, NE, USA). For each seedling, three leaves were selected, with five replicate measurements per leaf. The system’s airflow rate was set to 400 μmol/s, and the photosynthetic active radiation (PAR) intensity was maintained at 1000 µmol/m^2^/s.

### 4.3. Determination of Physiological and Biochemical Indicators

The activities of peroxidase (POD), superoxide dismutase (SOD) and catalase (CAT), together with the contents of malondialdehyde (MDA), proline (Pro), soluble sugars(SS) and soluble proteins (SP), were determined using commercial 96-well microplate assay kits (Quanzhou Ruixin Biotechnology Co., Ltd., Quanzhou, China). The following kits were used: Superoxide Dismutase (SOD) Activity Assay Kit (WST-8 method; Cat. No. G0101W), Peroxidase (POD) Activity Assay Kit (Cat. No. G0107W), Catalase (CAT) Activity Assay Kit (Cat. No. G0105W), Malondialdehyde (MDA) Content Assay Kit (Cat. No. G0109W), Proline (Pro) Content Assay Kit (Cat. No. G0111W), Soluble Sugar Content Assay Kit (Cat. No. G0501W) and Soluble Protein Content Assay Kit (Coomassie Brilliant Blue method; Cat. No. G0417W). All procedures followed manufacturer protocols with three technical replicates per sample.

### 4.4. Membership Function Analysis

Membership function analysis is a method derived from fuzzy mathematics [52]. Its principle lies in constructing membership functions to describe the fuzzy relationship between objects and specific characteristics. When applied to evaluate drought resistance in *C. oleifera*, the membership function of each physiological/biochemical indicator can be correlated with drought resistance capacity. The calculation of membership function values referenced the method proposed by Zhuo [53]. First, the membership function values for each indicator under different combinations were calculated. Then, the membership function values of all indicators within each group were summed and averaged to obtain the average value (D), which was used to evaluate the drought resistance of different combinations. The drought resistance of each combination was ranked based on the magnitude of its D value. The formula for calculating the membership function values is shown below:(1)$$u\left(X_{ij}\right)=\frac{X_{ij}-X_{jmin}}{X_{max}-X_{min}}$$(2)$$u\left(X_{ij}\right)=1-\frac{X_{ij}-X_{jmin}}{X_{jmax}-X_{jmin}}$$

In the formulas, u(X*_ij_*) represents the membership function value for the *j*-th indicator in the *i*-th combination. X*_ij_* is the measured mean value of the *j*-th indicator in the *i*-th combination. X*_jmax_* and X*_jmin_* denote the maximum and minimum values, respectively, of the *j*-th indicator across all combinations. Formula (1) was applied when an indicator was positively correlated with drought resistance, while Formula (2) was used for indicators negatively correlated with drought resistance. Membership function values were computed using Microsoft Excel 2021 Standard Edition.

### 4.5. RNA Extraction and Real-Time Quantitative PCR (qRT-PCR)

Total RNA was extracted from *C. oleifera* leaves using the RNAprep Pyre Plant Plus Kit (TIANGEN, Beijing, China). Immediately after extraction, first-strand cDNA was synthesized using the HiScript III 1st Strand cDNA Synthesis Kit (+gDNA wiper) (Vazyme, Nanjing, China). The resulting cDNA was diluted 5-fold and used as the template for qRT-PCR. *CoGAPDH* was used as the reference gene due to its validated stability in *C. oleifera* under drought stress. To confirm stability in our experiment, we analyzed *CoGAPDH* CT values using geNorm, supporting its reliability as a single reference gene. ChamQ Universal SYBR qPCR Master Mix (Vazyme, Nanjing, China) was used for the qRT-PCR reactions. Primers were designed using Primer Premier 5 software (Premier Biosoft, Palo Alto, CA, USA); their sequences are listed in Table 3. The relative expression levels of target genes were calculated using the 2^−∆∆Ct^ method.

### 4.6. Statistical Analysis

Data were collected, organized, and preliminarily analyzed using Microsoft Excel 2021 Standard Edition. Membership function analysis was also performed within Excel. Figures were generated using GraphPad Prism software 10.5.0. To assess significant differences in photosynthetic parameters and physiological indicators among the different combinations, one-way analysis of variance (ANOVA) was performed, followed by Duncan’s multiple range test for post-hoc comparisons. Statistical models accounted for hierarchical replication: biological replicates (n = 3) as random effects and graft/drought treatments as fixed effects. Analytical replicates (photosynthesis: n = 5 readings/leaf; biochemistry/qRT-PCR: n = 3 technical replicates/sample) were averaged prior to statistical testing.

## 5. Conclusions

Under drought stress, grafted *Camellia oleifera* seedlings reduced stomatal conductance and transpiration while accumulating osmolytes (proline, soluble sugars) and increasing antioxidant enzyme activity to conserve water and mitigate oxidative damage. Drought tolerance varied significantly among graft combinations, with combination C5 exhibiting superior water use efficiency, higher antioxidant activity, and lower cellular oxidation. Membership function analysis identified the graft combination using ‘Xianglin210’ scion and “Xianglin27” rootstock as the most drought-tolerant. Drought stress also upregulated key drought-responsive genes (*CoSnRK2.8*, *PP2Cs*), activating the ABA signaling pathway.

## Figures and Tables

**Figure 1 plants-14-02568-f001:**
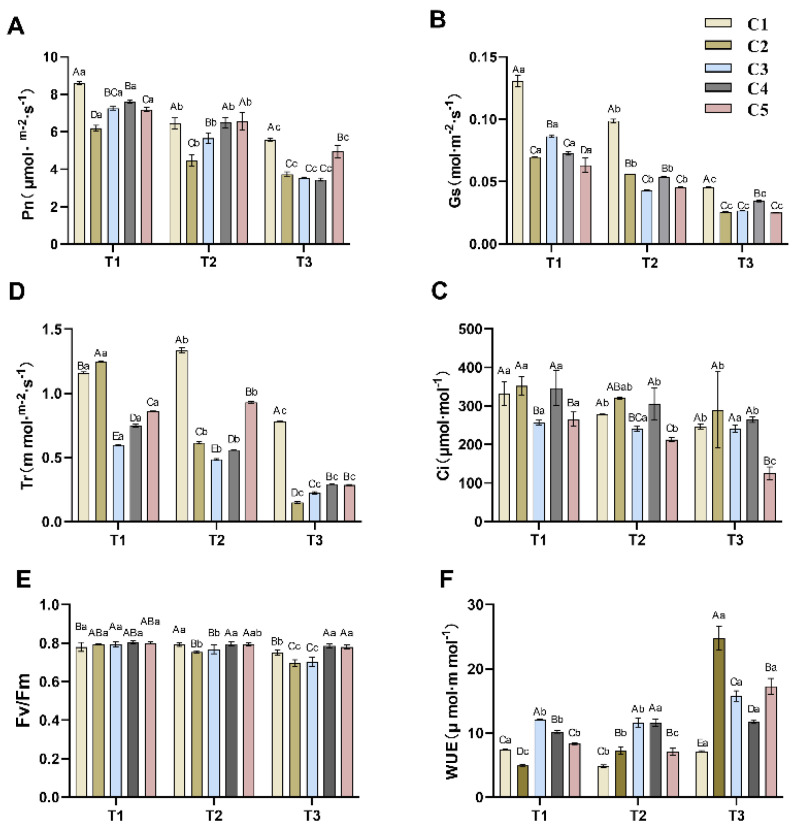
Effects of different soil moisture content on five different rootstock *C. oleifera* grafting combinations Pn (**A**), Gs (**B**), Ci (**C**), Tr (**D**), Fv/Fm (**E**), and WUE (**F**). Different uppercase letters indicate significant differences in the mean values of different combinations under the same soil moisture content (*p* < 0.05), and different lowercase letters indicate significant differences in the mean values of the same combination under different soil moisture contents (*p* < 0.05). T1, T2, T3—soil moisture treatments corresponding to 6, 8, and 10 days of water withholding, respectively; C1, C2, C3, C4, C5—different rootstock–scion grafting combinations (see Table 1 for details). Pn—net photosynthetic rate; Gs—stomatal conductance; Ci—intercellular CO_2_ concentration; Tr—transpiration rate; WUE—water use efficiency; Fv/Fm—theoretical maximum photochemical efficiency.

**Figure 2 plants-14-02568-f002:**
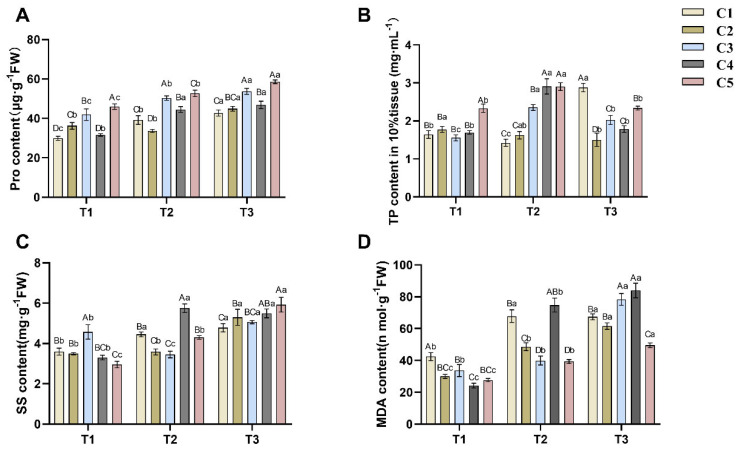
Effects of different soil moisture content on five different rootstock C.a oleifera grafting combinations: Pro (**A**), TP (**B**), SS (**C**), and MDA (**D**). Different uppercase letters indicated significant differences in the mean values of different combinations under the same soil moisture content (*p* < 0.05), and different lowercase letters indicated significant differences in the mean values of the same combinations under different soil moisture contents (*p* < 0.05). T1, T2, T3—Soil moisture treatments corresponding to 6, 8, and 10 days of water withholding, respectively; C1, C2, C3, C4, C5—different rootstock–scion grafting combinations (see Table 1 for details). MDA—malondialdehyde; Pro—proline; SS—soluble sugars; TP—total phosphorus.

**Figure 3 plants-14-02568-f003:**
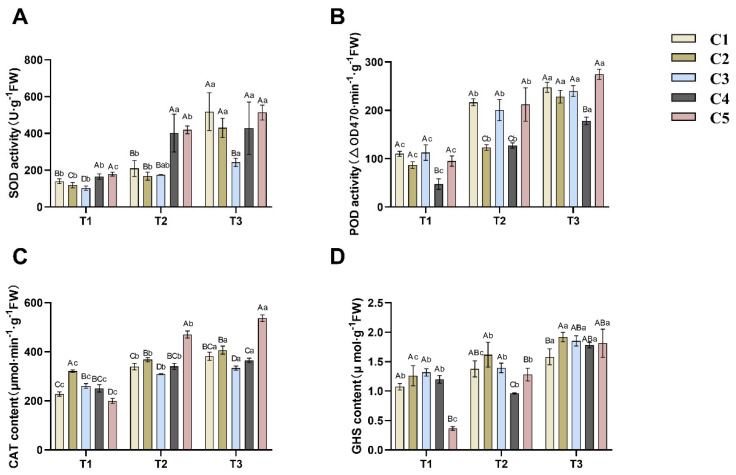
Effects of different soil moisture content on five different rootstock *C. oleifera* grafting combinations: SOD (**A**), POD (**B**), CAT (**C**), and GSH (**D**). Different uppercase letters indicated significant differences in the mean values of different combinations under the same soil moisture content (*p* < 0.05), and different lowercase letters indicated significant differences in the mean values of the same combinations under different soil moisture contents (*p* < 0.05). T1, T2, T3—soil moisture treatments corresponding to 6, 8, and 10 days of water withholding, respectively; C1, C2, C3, C4, C5—different rootstock–scion grafting combinations (see Table 1 for details). SOD—superoxide dismutase; POD—peroxidase; CAT—catalase; GSH—reduced glutathione.

**Figure 4 plants-14-02568-f004:**
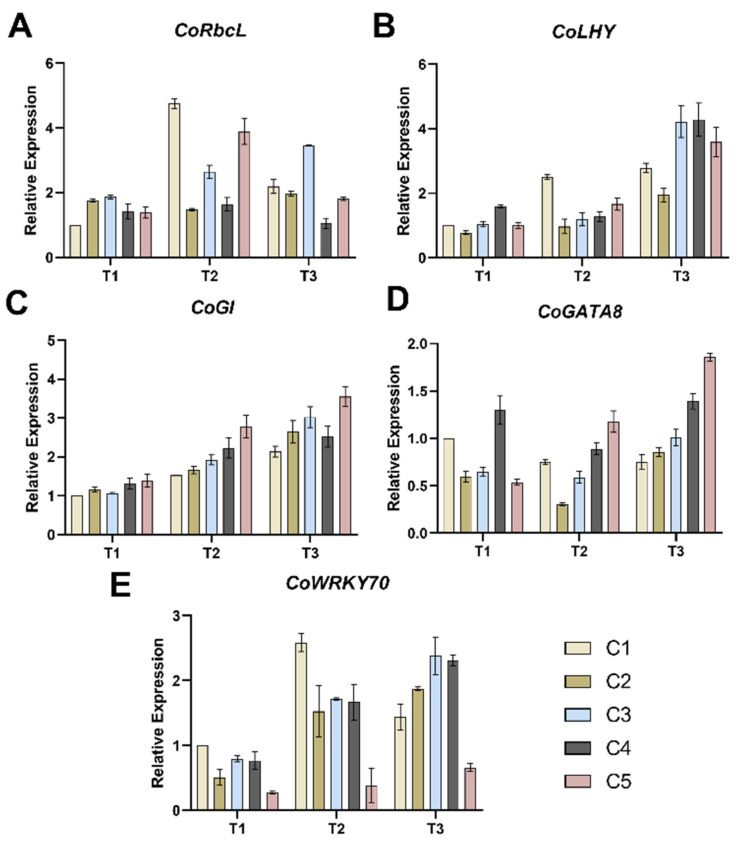
Effects of different soil moisture contents on the expression levels of five combinations of *CoRbcL* (**A**), *CoLHY* (**B**), *CoGI* (**C**), *CoGATA8* (**D**) and *CoWRKY70* (**E**). Vertical bars indicate standard deviation of the mean (n = 3). T1, T2, T3—soil moisture treatments corresponding to 6, 8, and 10 days of water withholding, respectively; C1, C2, C3, C4, C5—different rootstock–scion grafting combinations (see Table 1 for details).

**Figure 5 plants-14-02568-f005:**
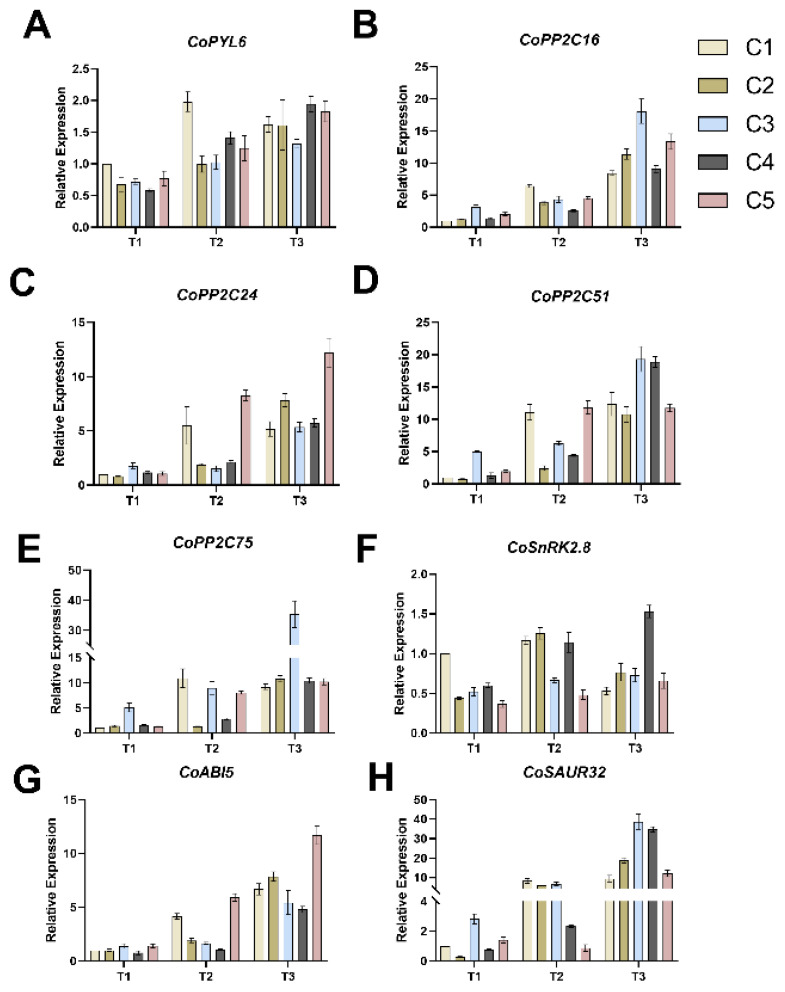
Effects of different soil water contents on the expression levels of *CoPYL6* (**A**), *CoPP2C16* (**B**), *CoPP2C24* (**C**), *CoPP2C51* (**D**), *CoPP2C75* (**E**), *CoSnRK2.8* (**F**), *CoABI5* (**G**), and *CoSAUR32* (**H**) in five combined grafting seedlings. Vertical bars indicate standard deviation of the mean (n = 3). T1, T2, T3—soil moisture treatments corresponding to 6, 8, and 10 days of water withholding, respectively; C1, C2, C3, C4, C5—different rootstock–scion grafting combinations (see Table 1 for details).

**Table 1 plants-14-02568-t001:** Comprehensive evaluation of drought resistance of different grafting combinations.

Group	Membership Function Value														D	Rank
	Pn	Gs	Ci	Tr	Fv/Fm	WUE	Pro	TP	SS	MDA	SOD	POD	CAT	GSH		
C1	1.00	1.00	0.73	0.00	0.39	0.00	0.00	1.00	0.00	0.48	1.00	0.72	0.24	0.00	0.47	3
C2	0.13	0.02	1.00	0.00	1.00	1.00	0.14	0.00	0.45	0.65	0.68	0.52	0.36	1.00	0.50	2
C3	0.04	0.07	0.70	0.12	0.92	0.49	0.70	0.38	0.25	0.16	0.00	0.64	0.00	0.81	0.38	4
C4	0.00	0.45	0.84	0.23	0.00	0.26	0.26	0.21	0.62	0.00	0.67	0.00	0.15	0.59	0.31	5
C5	0.70	0.00	0.00	0.21	0.06	0.57	1.00	0.61	1.00	1.00	0.98	1.00	1.00	0.69	0.63	1

**Table 2 plants-14-02568-t002:** Experiment information processing table.

Code	Rootstock	Rootstock Source	Scion	Treatment
C1	*C. yuhsienensis*	Changsha, Hunan Province; mother tree originated from seedling (12-year-old)	*Camellia oleifera* “Xianglin 210” (12-year-old)	T1 (plants subjected to water withholding for 6 days)
C1	*C. yuhsienensis*	Changsha, Hunan Province; mother tree originated from seedling (12-year-old)	*Camellia oleifera* “Xianglin 210” (12-year-old)	T2 (plants subjected to water withholding for 8 days)
C1	*C. yuhsienensis*	Changsha, Hunan Province; mother tree originated from seedling (12-year-old)	*Camellia oleifera* “Xianglin 210” (12-year-old)	T3 (plants subjected to water withholding for 10 days)
C2	*C. oleifera* (Guangxi Superior Germplasm)	Nanning, Guangxi Zhuang Autonomous Region; mother tree originated from seedling (10-year-old)	*Camellia oleifera* “Xianglin 210” (12-year-old)	T1 (plants subjected to water withholding for 6 days)
C2	*C. oleifera* (Guangxi Superior Germplasm)	Nanning, Guangxi Zhuang Autonomous Region; mother tree originated from seedling (10-year-old)	*Camellia oleifera* “Xianglin 210” (12-year-old)	T2 (plants subjected to water withholding for 8 days)
C2	*C. oleifera* (Guangxi Superior Germplasm)	Nanning, Guangxi Zhuang Autonomous Region; mother tree originated from seedling (10-year-old)	*Camellia oleifera* “Xianglin 210” (12-year-old)	T3 (plants subjected to water withholding for 10 days)
C3	*C. oleifera* (Hunan Superior Germplasm)	Changsha, Hunan Province; mother tree originated from tissue-cultured plant (12-year-old)	*Camellia oleifera* “Xianglin 210” (12-year-old)	T1 (plants subjected to water withholding for 6 days)
C3	*C. oleifera* (Hunan Superior Germplasm)	Changsha, Hunan Province; mother tree originated from tissue-cultured plant (12-year-old)	*Camellia oleifera* “Xianglin 210” (12-year-old)	T2 (plants subjected to water withholding for 8 days)
C3	*C. oleifera* (Hunan Superior Germplasm)	Changsha, Hunan Province; mother tree originated from tissue-cultured plant (12-year-old)	*Camellia oleifera* “Xianglin 210” (12-year-old)	T3 (plants subjected to water withholding for 10 days)
C4	*C. oleifera* ‘Xianglin1’	Changsha, Hunan Province; mother tree originated from grafted plant (18-year-old)	*Camellia oleifera* “Xianglin 210” (12-year-old)	T1 (plants subjected to water withholding for 6 days)
C4	*C. oleifera* ‘Xianglin1’	Changsha, Hunan Province; mother tree originated from grafted plant (18-year-old)	*Camellia oleifera* “Xianglin 210” (12-year-old)	T2 (plants subjected to water withholding for 8 days)
C4	*C. oleifera* ‘Xianglin1’	Changsha, Hunan Province; mother tree originated from grafted plant (18-year-old)	*Camellia oleifera* “Xianglin 210” (12-year-old)	T3 (plants subjected to water withholding for 10 days)
C5	*C. oleifera* ‘Xianglin27’	Changsha, Hunan Province; mother tree originated from grafted plant (18-year-old)	*Camellia oleifera* “Xianglin 210” (12-year-old)	T1 (plants subjected to water withholding for 6 days)
C5	*C. oleifera* ‘Xianglin27’	Changsha, Hunan Province; mother tree originated from grafted plant (18-year-old)	*Camellia oleifera* “Xianglin 210” (12-year-old)	T2 (plants subjected to water withholding for 8 days)
C5	*C. oleifera* ‘Xianglin27’	Changsha, Hunan Province; mother tree originated from grafted plant (18-year-old)	*Camellia oleifera* “Xianglin 210” (12-year-old)	T3 (plants subjected to water withholding for 10 days)

**Table 3 plants-14-02568-t003:** Gene primer sequences used for the quantitative real-time PCR analysis.

Gene	Forward Primer (5′–3′)	Reverse Primer (5′–3′)
*CoGAPDH*	CAGGTCGAGCATCTTTGATTCC	CCACCAACTTAACAAAGAAATCATTC
*CoRbcL*	TGGCATCCAAGTTGAAAGAG	ACGCATAAATGGTTGGGAGT
*CoLHY*	GAGGCGAAGCAGAGCAAGG	CACACATCCACACGACAAGTTTC
*CoGI*	GTTGGTGTGGTGGAGTGATGG	CCGTTGTTGGAGGAGGAAGC
*CoGATA8*	GTAGCAGCAGCAGCAGTTC	TCGGTGATGGAAGAGGTTGG
*CoWRKY70*	GATGAGTTCTTGTTGTGATGAGTC	GCACCTAAAGTAGCACCTTGG
*CoPYL6*	ACTCTTCAATCATCACTGTCCATCC	CTGCTCCAACAAATGTCTACAAAGG
*CoPP2C16*	CTACGGTGGCAGTGAATAGTG	CTTCCATCTCTGACCTCTTTCC
*CoPP2C24*	TCTGATACTGGCGAGCGATG	CCACCACAACCACACTTACG
*CoPP2C51*	AGAAGCCTGATAGAGAAGATGAAC	ATCCTCGTCACTCCTTGTCG
*CoPP2C75*	GCTATTGACTGCTCTGGTTGTTC	TTGCGGTTTCGTGTTCTATTCG
*CoSnRK2.8*	TTCGGCTACTCAAAGTCATCAG	CACCAACCAACATCACATATAAGG
*CoABI5*	GATGACACTGGAGGAGTTCTTG	TTGTTCTCTGGTATCCGATTAGC
*CoSAUR32*	CCACCACCACCACCATCAC	CTTCTCATCAGCATCGTCTACAAC

## Data Availability

The data supporting the findings of this study are available from the corresponding author upon reasonable request via email at wangrui102@163.com.

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
