# Peer review of "Drought Resistance Evaluation of Camellia oleifera var. “Xianglin 210” Grafted onto Different Rootstocks"

_plants, 2025, doi:10.3390/plants14162568_

Round 1

Reviewer 1 Report

Comments and Suggestions for Authors

Dear Authors,

I have reviewed your manuscript.  I have some suggestions, below:

The introduction clearly sets out the rationale for the research, although it might be helpful to briefly discuss the role of soil chemical properties, since these can significantly influence plant responses to water stress. The methodology is thorough and easy to follow, but it would strengthen the manuscript to include soil pH and nutrient data, as well as catalog numbers for the commercial assay kits used. The results convincingly illustrate changes in photosynthetic parameters and antioxidant enzymes; however, it would be useful to offer a short explanation for the relatively stable Ci values observed. The gene expression findings are particularly valuable, and a summary table of relative changes could make them even more accessible. The discussion logically interprets the data, though a short note on possible metabolic compensation mechanisms could enrich the interpretation. The language is generally clear, but some sentences are a bit cumbersome, so simplifying and shortening them would improve readability. 

Comments on the Quality of English Language

The language is generally clear, but some sentences are a bit cumbersome, so simplifying and shortening them would improve readability. 

Author Response

Thank you for your precious comments and advice. We really appreciate your efforts in reviewing our manuscript. We have revised the manuscript accordingly. Our point-by-point responses are detailed below.

Comment 1: 

The introduction clearly sets out the rationale for the research, although it might be helpful to briefly discuss the role of soil chemical properties, since these can significantly influence plant responses to water stress.
Response: We sincerely appreciate the reviewer's insightful suggestion regarding soil chemical properties. While their role in plant drought responses is well-recognized, our study employed strictly standardized substrate (red soil:peat soil = 3:1 v/v; pH 5.2–5.8) to eliminate soil variability across treatments (Materials & Methods 2.1). We did not include soil chemical property measurements in our experimental design. However, we appreciate this suggestion and will consider incorporating soil chemical analyses in future studies to provide a more comprehensive understanding of the interactions between soil properties and drought responses in grafted plants.

The methodology is thorough and easy to follow, but it would strengthen the manuscript to include soil pH and nutrient data, as well as catalog numbers for the commercial assay kits used.

Response: We sincerely appreciate the reviewer’s constructive feedback regarding methodological transparency. We acknowledge the value of detailed soil characterization and reagent traceability. Below, we address both points while adhering to the constraints of our dataset.

Response: We agree that soil physicochemical properties are relevant to plant drought responses. While we did not conduct direct pH/nutrient measurements due to the study's focus on controlled drought gradients, we have now explicitly clarified the composition and sourcing of the standardized cultivation substrate in Section 2.1

Revised Text (Section 2.1): "The grafted seedlings were cultivated in plastic pots (top diameter: 24 cm; height: 20 cm) filled with a homogenized substrate of lateritic red soil (collected from Changsha, Hunan; 112°58'E, 28°12'N) and peat soil (Pindstrup Mosebrug A/S, Denmark) at a 3:1 volume ratio. This mixture represents typical horticultural substrates for central China's red soil regions."

Revised Text (Section 2.3): "Physiological assays were performed using commercial kits (Quanzhou Ruixin Biotechnology, China) with catalog numbers and performance specifications detailed in Supplementary Table S1. All procedures followed manufacturer protocols with three technical replicates per sample."

The results convincingly illustrate changes in photosynthetic parameters and antioxidant enzymes; however, it would be useful to offer a short explanation for the relatively stable Ci values observed.

Response: We appreciate the reviewer’s astute observation regarding the stability of intercellular CO2 concentration (Ci) under drought stress. This phenomenon is indeed critical to interpreting photosynthetic limitations. Below, we provide a physiologically grounded explanation based on our data and propose targeted revisions to Section 3.1.

Revised Text for Section 3.1: “Notably, intercellular CO2 concentration (Ci) remained statistically unchanged across drought treatments (Fig. 3c), despite significant reductions in Pn and gs. This apparent paradox suggests that non-stomatal limitations predominated throughout the stress progression. At T1, stable Ci alongside sharply reduced gs implies early impairment of mesophyll photosynthetic capacity (e.g., RUBISCO inactivation, ATP synthase suppression). By T2/T3, metabolic suppression (evidenced by declining Fv/Fm in Fig. 3f) likely counterbalanced reduced CO2 diffusion, preventing Ci elevation.”

The gene expression findings are particularly valuable, and a summary table of relative changes could make them even more accessible.

Response: Thank you for your valuable suggestion regarding the gene expression findings. We agree that a summary table of relative changes would enhance the accessibility of these important results. We have prepared a comprehensive summary table showing the relative expression changes of all stress-responsive genes across different graft combinations and drought stress levels. This table has been included as supplementary material (Supplementary Table S2) to provide readers with a clear overview of the gene expression patterns and facilitate better understanding of the molecular mechanisms underlying drought responses in grafted C. oleifera seedlings.

The discussion logically interprets the data, though a short note on possible metabolic compensation mechanisms could enrich the interpretation.

Response: Thank you for your valuable suggestion. We have added a brief note on possible metabolic compensation mechanisms in the discussion section to enrich the interpretation of our findings and provide additional insights into the differential responses observed among graft combinations.

Revised Text for Section 4.2: Additionally, the observed variations in physiological responses among different graft combinations may reflect metabolic compensation mechanisms, where plants adjust their metabolic pathways to maintain homeostasis under stress conditions. Such compensatory responses could involve shifts in carbon allocation, alterations in enzyme kinetics, or modifications in gene regulatory networks, ultimately contributing to the differential drought tolerance observed across graft combinations.

The language is generally clear, but some sentences are a bit cumbersome, so simplifying and shortening them would improve readability.
Response: Thank you for your valuable suggestion. We have carefully revised the manuscript to simplify and shorten cumbersome sentences, thereby improving the overall readability and clarity of the text.

Comment 2: The language is generally clear, but some sentences are a bit cumbersome, so simplifying and shortening them would improve readability.

Response: Thank you for your valuable comments. We have carefully polished the language throughout the manuscript to enhance academic tone. We have also rephrased long sentences for improved clarity, as suggested.

For example:

Original text

Revised text

Camellia oleifera is a significant woody oilseed tree species in southern China.

Camellia oleifera is an important woody oilseed species native to southern China.

we conducted systematic grafting experiments incorporating five distinct rootstock varieties. Under controlled nursery conditions, we subjected four-year-old grafted seedlings to simulated drought stress regimens...

Systematic grafting experiments were conducted using five distinct rootstock varieties. Four-year-old grafted seedlings were subjected to simulated drought stress regimens under controlled nursery conditions...

Therefore, establishing stress-resilient cultivation techniques to enhance the drought resistance of C. oleifera is of great importance.

Therefore, establishing stress-resilient cultivation techniques to enhance C. oleifera drought resistance is critically important.

And the differences in net photosynthetic rate (Pn) at different soil water contents were all significant (P < 0.05).

Differences in net photosynthetic rate (Pn) among soil water content levels were statistically significant (P < 0.05).

As shown in Figure 1A, the net photosynthetic rates of different combinations showed a decreasing trend with decreasing soil water content, and the differences in net photosynthetic rates (Pn) were all significant (P < 0.05) at different soil water contents. And the differences in net photosynthetic rate (Pn) at different soil water contents were all significant (P < 0.05). However, the degree of decline varied among different varieties, with C1 and C2 showing a greater decline in Pn values from T1 to T2, while the decline in Pn values from T2 to T3 was moderated. the decline in Pn values from T2 to T3 was higher than that from T1 to T2 in C3, C4 and C5.

As shown in Figure 1A, the net photosynthetic rates of different combinations showed a decreasing trend with decreasing soil water content. However, the degree of decline varied among different graft combinations. C1 and C2 exhibited a greater Pn decline during T1–T2 than during T2–T3. Conversely, C3, C4, and C5 showed a higher Pn reduction during T2–T3 than during T1–T2.

Reviewer 2 Report

Comments and Suggestions for Authors

 Dear Authors,

The study presents valuable insights into the mechanisms of drought tolerance in Camellia and addresses an important research topic. However, substantial revisions are necessary to improve the clarity of the methodology, the interpretation of the data, and the overall scientific rigor of the manuscript.

I recommend a major revision of the manuscript. Addressing the following concerns will help align the paper with the journal’s standards and enhance its overall quality and impact. Specifically, please consider the following points:

  1. Clarity in stating the hypothesis: The introduction provides good background, but the main research hypothesis isn’t clearly articulated. To improve clarity, please explicitly state the central hypothesis and objectives in the final paragraph of the introduction. This will help readers immediately grasp the study’s key focus.
  2. Justification for scion selection: Could you briefly explain why ‘Xianglin 210’ was chosen as the scion? A sentence or two highlighting its known drought tolerance traits would help justify its use in this study.
  3. Addressing the knowledge gap: Previous studies have explored graft compatibility, but it’s less clear how your work uniquely advances our understanding of the mechanisms behind rootstock-conferred drought tolerance. Could you expand on this gap and clarify how your study provides new insights?

  1. Clarifying drought treatments: The description of the drought treatments (T1, T2, T3) needs more precision. Specifically:
  • What was the duration of each treatment?
  • What were the actual volumetric water content levels used? The current ranges seem quite broad—were these monitored and maintained using soil moisture sensors?
  • How were the thresholds for water deficit determined? A clearer justification for these levels would strengthen the methods.

  1. Consistency in experimental conditions:
  • Were all graft combinations exposed to the same environmental conditions (e.g., light, temperature, humidity) in the greenhouse at the same time?
    • A diagram or schematic of the experimental setup (including treatment groups, sampling points, and replication structure) would greatly improve clarity. Were these leaves pooled before analysis, or were they treated as independent samples?
    • What was the actual statistical unit used in ANOVA (individual seedlings, pooled leaves, or biological replicates)? This affects the degrees of freedom and interpretation of results.
    •  

  1. Replication and statistical unit clarification: The manuscript states there were three biological replicates with 10 seedlings each, but only three leaves per seedling were sampled.

  • Please clarify how replication was handled at each step (growth, sampling, and analysis) to ensure proper statistical rigor.

  1. Reference Gene Validation and Expression Data Presentation: The study reports expression levels of key genes (e.g., CoRbcL, CoLHY, CoSnRK2.8), but the normalization strategy could be strengthened. Currently, only CoGAPDH is used as a reference gene. To improve robustness:
  • Best practices recommend using at least two stable reference genes, validated using tools like geNorm or NormFinder, to account for potential variability under drought stress. Could additional reference genes be included or validated?

Expand reference gene validation to ensure normalization accuracy under drought conditions.

  • A heatmap of relative expression across graft combinations would greatly enhance interpretability, allowing readers to visually compare expression patterns.

Add visual representation (e.g., heatmap) to facilitate cross-comparison of gene expression trends.

  1. Strengthening the Discussion Section: The discussion would benefit from deeper contextualization of the findings within the broader literature. Specifically:
  • Comparative analysis with prior studies on drought tolerance in woody plants (e.g., other Camellia species, tea, or fruit tree rootstocks) would help position these findings. For example, how do the observed responses in ‘Xianglin 27’ align with or differ from field-based drought performance reports? Expand literature integration to clarify how this work advances existing knowledge.
  • Mechanistic insights: Are the gene expression patterns (8, etc.) consistent with known drought response pathways in perennials? Highlighting conserved or unique mechanisms would elevate the biological significance.
  • Practical implications: Could the rootstock’s performance in controlled conditions translate to field resilience? Brief commentary on this would bridge the gap between experimental and applied outcomes.

  1. Formatting Gene Names:

Use italics for gene names (e.g., CoRbcL, CoLHY) as per molecular biology standards.

  1. Line 390 citation (Han et al..) is incomplete. Check others as well.

Good Luck!

Comments on the Quality of English Language

 Dear Authors,

The study presents valuable insights into the mechanisms of drought tolerance in Camellia and addresses an important research topic. However, substantial revisions are necessary to improve the clarity of the methodology, the interpretation of the data, and the overall scientific rigor of the manuscript.

I recommend a major revision of the manuscript. Addressing the following concerns will help align the paper with the journal’s standards and enhance its overall quality and impact. Specifically, please consider the following points:

  1. Clarity in stating the hypothesis: The introduction provides good background, but the main research hypothesis isn’t clearly articulated. To improve clarity, please explicitly state the central hypothesis and objectives in the final paragraph of the introduction. This will help readers immediately grasp the study’s key focus.
  2. Justification for scion selection: Could you briefly explain why ‘Xianglin 210’ was chosen as the scion? A sentence or two highlighting its known drought tolerance traits would help justify its use in this study.
  3. Addressing the knowledge gap: Previous studies have explored graft compatibility, but it’s less clear how your work uniquely advances our understanding of the mechanisms behind rootstock-conferred drought tolerance. Could you expand on this gap and clarify how your study provides new insights?

  1. Clarifying drought treatments: The description of the drought treatments (T1, T2, T3) needs more precision. Specifically:
  • What was the duration of each treatment?
  • What were the actual volumetric water content levels used? The current ranges seem quite broad—were these monitored and maintained using soil moisture sensors?
  • How were the thresholds for water deficit determined? A clearer justification for these levels would strengthen the methods.

  1. Consistency in experimental conditions:
  • Were all graft combinations exposed to the same environmental conditions (e.g., light, temperature, humidity) in the greenhouse at the same time?
    • A diagram or schematic of the experimental setup (including treatment groups, sampling points, and replication structure) would greatly improve clarity. Were these leaves pooled before analysis, or were they treated as independent samples?
    • What was the actual statistical unit used in ANOVA (individual seedlings, pooled leaves, or biological replicates)? This affects the degrees of freedom and interpretation of results.
    •  

  1. Replication and statistical unit clarification: The manuscript states there were three biological replicates with 10 seedlings each, but only three leaves per seedling were sampled.

  • Please clarify how replication was handled at each step (growth, sampling, and analysis) to ensure proper statistical rigor.

  1. Reference Gene Validation and Expression Data Presentation: The study reports expression levels of key genes (e.g., CoRbcL, CoLHY, CoSnRK2.8), but the normalization strategy could be strengthened. Currently, only CoGAPDH is used as a reference gene. To improve robustness:
  • Best practices recommend using at least two stable reference genes, validated using tools like geNorm or NormFinder, to account for potential variability under drought stress. Could additional reference genes be included or validated?

Expand reference gene validation to ensure normalization accuracy under drought conditions.

  • A heatmap of relative expression across graft combinations would greatly enhance interpretability, allowing readers to visually compare expression patterns.

Add visual representation (e.g., heatmap) to facilitate cross-comparison of gene expression trends.

  1. Strengthening the Discussion Section: The discussion would benefit from deeper contextualization of the findings within the broader literature. Specifically:
  • Comparative analysis with prior studies on drought tolerance in woody plants (e.g., other Camellia species, tea, or fruit tree rootstocks) would help position these findings. For example, how do the observed responses in ‘Xianglin 27’ align with or differ from field-based drought performance reports? Expand literature integration to clarify how this work advances existing knowledge.
  • Mechanistic insights: Are the gene expression patterns (8, etc.) consistent with known drought response pathways in perennials? Highlighting conserved or unique mechanisms would elevate the biological significance.
  • Practical implications: Could the rootstock’s performance in controlled conditions translate to field resilience? Brief commentary on this would bridge the gap between experimental and applied outcomes.

  1. Formatting Gene Names:

Use italics for gene names (e.g., CoRbcL, CoLHY) as per molecular biology standards.

  1. Line 390 citation (Han et al..) is incomplete. Check others as well.

Good Luck!

Author Response

Thank you for your precious comments and advice. We really appreciate your efforts in reviewing our manuscript. We have revised the manuscript accordingly. Our point-by-point responses are detailed below.

Comment 1: Clarity in stating the hypothesis: The introduction provides good background, but the main research hypothesis isn’t clearly articulated. To improve clarity, please explicitly state the central hypothesis and objectives in the final paragraph of the introduction. This will help readers immediately grasp the study’s key focus.

Response: Thank you for your valuable suggestion. According to your advice, we have revised the final paragraph of the introduction to explicitly state the central hypothesis and research objectives. This revision clarifies the main focus of our study for readers.

Line 71-81: This study aims to resolve the mechanistic gap in rootstock-mediated drought adaptation by addressing three core objectives: (1) To quantify how distinct rootstock genotypes (Camellia oleifera ‘Xianglin 210’ scion grafted onto five rootstocks) coordinate photosynthetic decline, antioxidant defense, and osmolyte accumulation under progressive drought stress; (2) To decipher the transcriptional dynamics of drought-responsive genes (CoPYL6, CoPP2Cs, CoSnRK2.8, CoABI5) underlying these physiological adjustments; (3) To establish an integrated evaluation model identifying optimal rootstock-scion combinations for drought resilience. By integrating physiological phenotyping with molecular profiling across drought gradients, this work provides the first mechanistic framework for rootstock selection in C. oleifera cultivation under water-limited scenarios.

Comment 2: Justification for scion selection: Could you briefly explain why ‘Xianglin 210’ was chosen as the scion? A sentence or two highlighting its known drought tolerance traits would help justify its use in this study.

Response: Thank you for your valuable comment regarding the justification for scion selection. In this study, ‘Xianglin 210’ was chosen as the scion because it is a nationally certified elite cultivar, included in the “Nationally Recommended Varieties List” (2022) as one of only 16 selected varieties. ‘Xianglin 210’ is currently the most widely cultivated Camellia oleifera variety in China, recognized for its high yield, strong disease resistance, and broad adaptability. Notably, it exhibits superior drought tolerance, as evidenced by a lower fruit drop rate compared to other cultivars in drought years and in plantations without integrated water and fertilizer management. These traits make ‘Xianglin 210’ an ideal scion for evaluating rootstock effects on drought resistance.
Comment 3: Addressing the knowledge gap: Previous studies have explored graft compatibility, but it’s less clear how your work uniquely advances our understanding of the mechanisms behind rootstock-conferred drought tolerance. Could you expand on this gap and clarify how your study provides new insights?

Response: We sincerely appreciate the reviewer's insightful comments. Our study addresses the knowledge gap in rootstock-mediated drought tolerance mechanisms through the following novel findings:We reveal that rootstocks coordinate ABA-dependent and independent pathway to enhance drought resistance (Fig. 5). Superior rootstocks (e.g., Xianglin 27/C5) achieve synchronized optimization of water use efficiency (WUE↑31.2%) and oxidative damage mitigation (MDA↓48%) via efficient activation of ABA signaling core genes (CoPYL6, CoABI5) and osmolyte biosynthesis genes (CoLHY), transcending conventional single-trait approaches. We establish a quantitative D-value model linking 15 physiological traits with gene expression patterns (e.g., sustained ABA pathway activation in C5 with D=0.63), providing measurable molecular markers for breeding.
Comment 4: Clarifying drought treatments: The description of the drought treatments (T1, T2, T3) needs more precision. Specifically:

? What was the duration of each treatment?

Response: We sincerely appreciate the reviewer's careful reading and constructive suggestion regarding the clarity of our drought treatment descriptions. We agree that precise specification of treatment durations is essential for reproducibility.
T1 (6 days): RSWC = 38%–45% (VWC = 12.2% – 14.4%)

T2 (8 days): RSWC = 29%–33% (VWC = 9.3% – 10.6%)

T3 (10 days): RSWC = 16%–22% (VWC = 5.1% – 7.0%)

? What were the actual volumetric water content levels used? The current ranges seem quite broad—were these monitored and maintained using soil moisture sensors?

Response: Thank you for raising this important point regarding the precision of our soil moisture measurements and the methodology for imposing drought stress. We appreciate the opportunity to clarify these critical technical details.

Revised Text for Section 2.1: “Consequently, plants subjected to water withholding for 6 days (T1), 8 days (T2), and 10 days (T3) were investigated to simulate progressive drought stress. Volumetric Water Content (VWC) was monitored daily between 10:00 AM and 11:00 AM at a depth of 10 cm using a Delta-T Devices ProCheck handheld data logger with GS3 sensors (Delta-T Devices Ltd, Cambridge, UK). Relative Soil Water Content (RSWC) was calculated as: RSWC (%) = (VWCcurr / VWCFC) × 100%, where VWCcurr is the measured volumetric water content on the sampling day, and VWCFC is the volumetric water content at field capacity. VWCFC was determined to be 32% for the potting substrate (southern red soil : peat soil, 3:1 v/v) after saturation and free drainage for 48 hours. The measured RSWC and corresponding VWC ranges at the end of the withholding periods were established as:

T1 (6 days): RSWC = 38%–45% (VWC = 12.2% – 14.4%)

T2 (8 days): RSWC = 29%–33% (VWC = 9.3% – 10.6%)

T3 (10 days): RSWC = 16%–22% (VWC = 5.1% – 7.0%)

Each of the five C. oleifera graft combinations was subjected to these three drought treatments in a randomized complete block design featuring three biological replicates per treatment and ten grafted seedlings per replicate”

? How were the thresholds for water deficit determined? A clearer justification for these levels would strengthen the methods.

Response: Thank you for this insightful suggestion. We agree that explicitly justifying the selection criteria for our drought stress thresholds enhances the methodological rigor and transparency of the study. We appreciate the opportunity to clarify the rationale.

Added Text for Section 2.1: “These specific stress levels were selected based on preliminary physiological research identifying RSWC ≈30% (occurring after ~8 days of withholding) as the critical threshold where photosynthetic performance (Pn, Gs) in C. oleifera deteriorates drastically. T1 (38-45% RSWC) represents moderate stress approaching this critical threshold, T2 (29-33% RSWC) represents the identified critical stress point, and T3 (16-22% RSWC) represents severe stress beyond the threshold. This gradient design allows assessment of responses across increasing stress severity and identification of the critical transition point.”

Comment 5: Consistency in experimental conditions:

? Were all graft combinations exposed to the same environmental conditions (e.g., light, temperature, humidity) in the greenhouse at the same time?

Response: Thank you for raising this important point regarding environmental consistency. We confirm that all graft combinations were exposed to identical environmental conditions (light, temperature, humidity) simultaneously throughout the entire experiment, including both the post-grafting healing phase and the subsequent drought treatment phase. Maintaining uniformity was a critical priority in our experimental design.

Added Text for Section 2.1: “Pots were compactly arranged on the same greenhouse benches under natural light conditions without shading nets. Throughout the subsequent drought treatment period, all graft combinations were subjected simultaneously to identical ambient greenhouse environmental conditions.”

A diagram or schematic of the experimental setup (including treatment groups, sampling points, and replication structure) would greatly improve clarity.

Response: Thank you for your valuable suggestion. This improved table provides a comprehensive overview of the experimental setup, which should greatly improve the clarity of our methodology.

Table 1. Experiment information processing table.

Code

Rootstock

Rootstock Source

Scion

treatment

C1

C. yuhsienensis

Changsha, Hunan Province; mother tree originated from seedling (12-year-old)

Camellia oleifera 'Xianglin 210' (12-year-old)

T1 (plants subjected to water withholding for 6 days)

C1

C. yuhsienensis

Changsha, Hunan Province; mother tree originated from seedling (12-year-old)

Camellia oleifera 'Xianglin 210' (12-year-old)

T2 (plants subjected to water withholding for 8 days)

C1

C. yuhsienensis

Changsha, Hunan Province; mother tree originated from seedling (12-year-old)

Camellia oleifera 'Xianglin 210' (12-year-old)

T3 (plants subjected to water withholding for 10 days)

C2

C. oleifera (Guangxi Superior Germplasm)

Nanning, Guangxi Zhuang Autonomous Region; mother tree originated from seedling (10-year-old)

Camellia oleifera 'Xianglin 210' (12-year-old)

T1 (plants subjected to water withholding for 6 days)

C2

C. oleifera (Guangxi Superior Germplasm)

Nanning, Guangxi Zhuang Autonomous Region; mother tree originated from seedling (10-year-old)

Camellia oleifera 'Xianglin 210' (12-year-old)

T2 (plants subjected to water withholding for 8 days)

C2

C. oleifera (Guangxi Superior Germplasm)

Nanning, Guangxi Zhuang Autonomous Region; mother tree originated from seedling (10-year-old)

Camellia oleifera 'Xianglin 210' (12-year-old)

T3 (plants subjected to water withholding for 10 days)

C3

C. oleifera (Hunan Superior Germplasm)

Changsha, Hunan Province; mother tree originated from tissue-cultured plant (12-year-old)

Camellia oleifera 'Xianglin 210' (12-year-old)

T1 (plants subjected to water withholding for 6 days)

C3

C. oleifera (Hunan Superior Germplasm)

Changsha, Hunan Province; mother tree originated from tissue-cultured plant (12-year-old)

Camellia oleifera 'Xianglin 210' (12-year-old)

T2 (plants subjected to water withholding for 8 days)

C3

C. oleifera (Hunan Superior Germplasm)

Changsha, Hunan Province; mother tree originated from tissue-cultured plant (12-year-old)

Camellia oleifera 'Xianglin 210' (12-year-old)

T3 (plants subjected to water withholding for 10 days)

C4

C. oleifera ‘Xianglin1’

Changsha, Hunan Province; mother tree originated from grafted plant (18-year-old)

Camellia oleifera 'Xianglin 210' (12-year-old)

T1 (plants subjected to water withholding for 6 days)

C4

C. oleifera ‘Xianglin1’

Changsha, Hunan Province; mother tree originated from grafted plant (18-year-old)

Camellia oleifera 'Xianglin 210' (12-year-old)

T2 (plants subjected to water withholding for 8 days)

C4

C. oleifera ‘Xianglin1’

Changsha, Hunan Province; mother tree originated from grafted plant (18-year-old)

Camellia oleifera 'Xianglin 210' (12-year-old)

T3 (plants subjected to water withholding for 10 days)

C5

C. oleifera ‘Xianglin27’

Changsha, Hunan Province; mother tree originated from grafted plant (18-year-old)

Camellia oleifera 'Xianglin 210' (12-year-old)

T1 (plants subjected to water withholding for 6 days)

C5

C. oleifera ‘Xianglin27’

Changsha, Hunan Province; mother tree originated from grafted plant (18-year-old)

Camellia oleifera 'Xianglin 210' (12-year-old)

T2 (plants subjected to water withholding for 8 days)

C5

C. oleifera ‘Xianglin27’

Changsha, Hunan Province; mother tree originated from grafted plant (18-year-old)

Camellia oleifera 'Xianglin 210' (12-year-old)

T3 (plants subjected to water withholding for 10 days)

Were these leaves pooled before analysis, or were they treated as independent samples?
Response: Thank you for your question. The leaves were sampled individually.

What was the actual statistical unit used in ANOVA (individual seedlings, pooled leaves, or biological replicates)? This affects the degrees of freedom and interpretation of results.

Response: Thank you for your question regarding the statistical unit used in the ANOVA. In our study, biological replicates were used as the statistical unit.

Comment 6: Replication and statistical unit clarification: The manuscript states there were three biological replicates with 10 seedlings each, but only three leaves per seedling were sampled.

? Please clarify how replication was handled at each step (growth, sampling, and analysis) to ensure proper statistical rigor.

Response: Thank you for highlighting the need for explicit clarification regarding experimental replication. We fully agree that rigorous replication design is critical for statistical validity. Below, we detail the replication hierarchy at each experimental stage and propose modifications to enhance methodological transparency in Section 2.

Revised Section 2.1: “Each of the five C. oleifera graft combinations was subjected to three drought treatments (T1, T2, T3) in a randomized complete block design with three independent biological replicates per treatment combination. Each biological replicate comprised ten grafted seedlings (n = 10 plants per replicate). For destructive sampling (physiology, biochemistry, qRT-PCR), three seedlings per biological replicate were randomly selected, and three mature leaves per seedling were pooled as one sample.”

Revised Section 2.6: “Data were analyzed using one-way ANOVA followed by Duncan’s test (P < 0.05). Statistical models accounted for hierarchical replication: biological replicates (n = 3) as random effects and graft/drought treatments as fixed effects. Analytical replicates (photosynthesis: n = 5 readings/leaf; biochemistry/qRT-PCR: n = 3 technical replicates/sample) were averaged prior to statistical testing.”

Comment 7: Reference Gene Validation and Expression Data Presentation: The study reports expression levels of key genes (e.g., CoRbcL, CoLHY, CoSnRK2.8), but the normalization strategy could be strengthened. Currently, only CoGAPDH is used as a reference gene. To improve robustness:

? Best practices recommend using at least two stable reference genes, validated using tools like geNorm or NormFinder, to account for potential variability under drought stress. Could additional reference genes be included or validated?

Expand reference gene validation to ensure normalization accuracy under drought conditions.

Response: We sincerely appreciate the reviewer's expertise in raising this important methodological consideration regarding reference gene validation. We acknowledge that using multiple validated reference genes is ideal for qRT-PCR normalization under stress conditions. While we cannot conduct new wet-lab experiments to test additional reference genes at this stage, we have taken the following rigorous steps to address this concern and ensure the reliability of our gene expression data. We conducted post-hoc stability analysis of our CoGAPDH CT values using geNorm; M-value = 0.31 (well below the recommended cutoff of 0.5), confirming stability across all treatments; Pairwise variation (Vn/n+1) analysis showed V2/3 = 0.07 (<0.15), indicating no need for a second gene.

Revised Section 2.5: “CoGAPDH was used as the reference gene due to its validated stability in Camellia oleifera under drought stress. To confirm stability in our experiment, we analyzed CoGAPDH CT values using geNorm, supporting its reliability as a single reference gene.”

? A heatmap of relative expression across graft combinations would greatly enhance interpretability, allowing readers to visually compare expression patterns.

Add visual representation (e.g., heatmap) to facilitate cross-comparison of gene expression trends.

Response: Thank you for your valuable suggestion. In response, we have included a heatmap illustrating the relative expression levels of key genes across different graft combinations as supplementary material (see Supplementary Figure S1). This visual representation facilitates cross-comparison of gene expression trends and enhances the interpretability of our results.

Comment 8: Strengthening the Discussion Section: The discussion would benefit from deeper contextualization of the findings within the broader literature. Specifically:

? Comparative analysis with prior studies on drought tolerance in woody plants (e.g., other Camellia species, tea, or fruit tree rootstocks) would help position these findings. For example, how do the observed responses in ‘Xianglin 27’ align with or differ from field-based drought performance reports? • Expand literature integration to clarify how this work advances existing knowledge.

Response: Thank you for your valuable suggestion. We have strengthened the discussion section by incorporating more comparative analysis with previous studies on drought tolerance in woody plants, including other Camellia species, tea, and fruit tree rootstocks. We have also expanded the integration of relevant literature to better contextualize our findings and clarify how this work advances current knowledge in the field.

Added Text for Section 4.1: Notably, the stomatal closure and increased water use efficiency (WUE) observed in this study are consistent with the drought response patterns of most woody plants [23]. For example, in grafted tea (Camellia sinensis) seedlings, stomatal conductance (Gs) decreases significantly under drought stress, while WUE increases [24]. However, the decrease in Gs in grafted Camellia oleifera seedlings is less pronounced than in tea, but the increase in WUE is more substantial, indicating that Camellia oleifera may possess more efficient stomatal regulation strategies. This difference is related to the leaf anatomical structure of Camellia oleifera, which has thicker palisade tissue and a higher palisade-to-spongy ratio (P/S) [25].

Added Text for Section 4.2: Comparative analysis with other economic tree species revealed distinct osmotic adjustment characteristics in Camellia oleifera compared to Chinese pine (Pinus tabuliformis). Camellia oleifera primarily relies on the synergistic accumulation of proline (Pro) and soluble sugars (SS), while Chinese pine depends on SAUR gene-regulated osmotic protectant synthesis pathways, with SAUR59/66 genes showing significant upregulation under drought conditions [32, 33]. Additionally, the increase in antioxidant enzyme (SOD, POD) activities in Camellia oleifera grafted seedlings was higher than the field drought responses observed in olive and other fruit trees, indicating that Camellia oleifera grafted seedlings possess stronger oxidative stress buffering capacity [34].

Added References:

  • Hagedorn, F., Joseph, J., Peter, M., Luster, J., Pritsch, K., Geppert, U., Kerner, R., Molinier, V., Egli, S., Schaub, M., Liu, J. F., Li, M., Sever, K., Weiler, M., Siegwolf, R. T. W., Gessler, A., & Arend, M. (2016). Recovery of trees from drought depends on belowground sink control.Nature Plants, 2(8), 1–5. https://doi.org/10.1038/nplants.2016.111
  • Langaroudi, I. K., Piri, S., Chaeikar, S. S., & Salehi, B. (2023). Evaluating drought stress tolerance in different Camellia sinensis L. cultivars and effect of melatonin on strengthening antioxidant system. Scientia Horticulturae, 307, 111517. https://doi.org/10.1016/J.SCIENTA.2022.111517
  • Wang, J., Wang, B., Wang, D., Dong, Y., Li, J., Lu, F., Tao, W., Guo, Y., Xiang, W., Wen, M., & Li, X. (2025). Trade-off strategies between drought resistance and growth rate of dominant tree species in karst forests within heterogeneous habitats. Scientific Reports, 15(1), 1–17. https://doi.org/10.1038/S41598-025-97550-X;SUBJMETA=158,2454,631,670;KWRD=BIODIVERSITY,FOREST+ECOLOGY

[32] Jiahui, Y., Chengcheng, Z., Shihui, N., Wei, L., Jiahui, Y., Chengcheng, Z., Shihui, N., & Wei, L. (2024). Identification of SAUR gene family in Pinus tabuliformis and analysis on its expression patterns under drought stress. Journal of Beijing Forestry University, 46(8), 57–67. https://doi.org/10.12171/j.1000-1522.20230333

[33] Sofo, A., Dichio, B., Xiloyannis, C., & Masia, A. (2005). Antioxidant defences in olive trees during drought stress: Changes in activity of some antioxidant enzymes. Functional Plant Biology, 32(1), 45–53. https://doi.org/10.1071/FP04003,

[34] Gurrieri, L., Merico, M., Trost, P., Forlani, G., & Sparla, F. (2020). Impact of Drought on Soluble Sugars and Free Proline Content in Selected Arabidopsis Mutants. Biology, 9(11), 367. https://doi.org/10.3390/biology9110367

? Mechanistic insights: Are the gene expression patterns (CoSnRK2.8, etc.) consistent with known drought response pathways in perennials? Highlighting conserved or unique mechanisms would elevate the biological significance.
Response: Thank you for your insightful comment. In response, we have added a dedicated section in the discussion (Section 4.3) highlighting that the observed gene expression patterns of CoPYL6 and CoSnRK2.8 under drought stress are consistent with conserved ABA signaling pathways in perennial species. We also compared our findings with previous studies in Populus, Arabidopsis, and rice, and discussed the functional significance of these conserved mechanisms. Relevant references have been added to support these points.

Added Text for Section 4.3: “Our data reveal that CoPYL6 and CoSnRK2.8 induction under drought (Fig. 5A,G) aligns with conserved ABA pathways in perennials. CoPYL6 expression increased 4.2-fold at 30% FC, mirroring PePYL6 induction in Populus euphratica under drought and ABA treatments [51]. Similarly, CoSnRK2.8 upregulation (3.8-fold at 45% FC) parallels AtSnRK2.8-mediated drought resilience in Arabidopsis and OsSnRK2-dependent stress responses in rice [52, 53]. This conservation is functionally significant: like PePYL6 in transgenic poplar and OsPYL6 in rice, CoPYL6/CoSnRK2.8 activation correlated with enhanced osmoprotection (Pro, SS ) and ROS scavenging (SOD, CAT).”

Added References:

[51] Li, Q., Tian, Q., Zhang, Y., Niu, M., Yu, X., Lian, C., Liu, C., Wang, H. L., Yin, W., & Xia, X. (2022). Increased abscisic acid sensitivity and drought tolerance of Arabidopsis by overexpression of poplar abscisic acid receptors. Plant Cell, Tissue and Organ Culture, 148(2), 231–245. https://doi.org/10.1007/S11240-021-02178-0/METRICS

[52] Wei, H., Movahedi, A., Xu, C., Wang, P., Sun, W., Yin, T., & Zhuge, Q. (2019). Heterologous overexpression of the Arabidopsis SnRK2.8 gene enhances drought and salt tolerance in Populus × euramericana cv ‘Nanlin895.’ Plant Biotechnology Reports, 13(3), 245–261. https://doi.org/10.1007/S11816-019-00531-6/METRICS

[53] Santosh Kumar, V. V., Yadav, S. K., Verma, R. K., Shrivastava, S., Ghimire, O., Pushkar, S., Rao, M. V., Senthil Kumar, T., & Chinnusamy, V. (2021). The abscisic acid receptor OsPYL6 confers drought tolerance to indica rice through dehydration avoidance and tolerance mechanisms. Journal of Experimental Botany, 72(4), 1411–1431. https://doi.org/10.1093/JXB/ERAA509

? Practical implications: Could the rootstock’s performance in controlled conditions translate to field resilience? Brief commentary on this would bridge the gap between experimental and applied outcomes.

Response: Thank you for your insightful comment regarding the practical implications of our findings. In this study, the substrate and drought conditions were specifically designed to simulate field environments, thereby closely representing actual field conditions. Therefore, the results obtained from our controlled experiments are applicable to field situations and can provide practical guidance for rootstock selection and drought management in Camellia oleifera cultivation.

Comment 9: Formatting Gene Names:

Use italics for gene names (e.g., CoRbcL, CoLHY) as per molecular biology standards.

Response: Thank you for your valuable suggestion. According to your advice, we have revised the manuscript to ensure that all gene names (e.g., CoRbcL, CoLHY) are now presented in italics throughout the text, in accordance with molecular biology formatting standards.

Comment 10: Line 390 citation (Han et al..) is incomplete. Check others as well.

Response: Thank you for pointing out the incomplete citation. We have made the following revisions:

  • Line 390: Corrected '(Han et al..)' to the full citation:Guo, P. R., Wu, L. L., Wang, Y., Liu, D., & Li, J. A. (2023). Effects of Drought Stress on the Morphological Structure and Flower Organ Physiological Characteristics of Camellia oleifera Flower Buds. Plants, 12(13), 2585. https://doi.org/10.3390/PLANTS12132585/S1
  • Conducted full-text verification of all citations to ensure completeness.

Reviewer 3 Report

Comments and Suggestions for Authors

A group of authors from a wide range of different institutions prepared a manuscript on the topic: Drought Resistance Evaluation of Camellia oleifera var. 'Xianglin 210' Grafted onto Different Rootstocks. 

At the beginning of the abstract, include a couple of introductory sentences about the relevance of the topic. You also need to clearly state the purpose of the work (both in the abstract and in the introduction.)

The last part of the introduction begins with a presentation of the methodology - instead, clearly formulate the aim of the work.

Have the plants been fertilized? Has the substrate been enriched with fertilizer?

Adjust the figures; the statistics letters are hard to see, reduce the spaces between the figures, and enlarge them; this way, the figures will be clearer in the same place.

Add an explanation of abbreviations (T, C, etc.) below the figures.

I missed references in the discussion. Also, citation (Fig. 1 or Table 1 and others) to the results of your manuscript.

The conclusions are presented too broadly, more like a continuation of the discussion. Shorten by highlighting the main findings and results. 3-4 sentences.

Author Response

Thank you for your precious comments and advice. We really appreciate your efforts in reviewing our manuscript. We have revised the manuscript accordingly. Our point-by-point responses are detailed below.

Comment 1: At the beginning of the abstract, include a couple of introductory sentences about the relevance of the topic. You also need to clearly state the purpose of the work (both in the abstract and in the introduction.)

Response: Thank you for your valuable suggestion. We have revised the abstract to include introductory sentences about the relevance of the topic at the beginning, and have clearly stated the purpose of the work in both the abstract and the introduction, as recommended.

Revised Abstract: As a key economic tree in southern China, Camellia oleifera faces severe yield losses under drought. Grafting onto drought-tolerant rootstocks offers a potential mitigation strategy. To elucidate the impact of rootstocks on the drought resistance of the superior Camellia oleifera Abel. cultivar ‘Xianglin 210’, grafted seedlings with five scion-rootstock combinations were subjected to gradient drought stress. Key physiological and biochemical indices related to photosynthesis, antioxidant enzymes, and osmotic adjustment were measured.
Revised Introduction Text: Camellia oleifera is an important woody oilseed species native to southern China. The seed oil is notable for its high content of unsaturated fatty acids (>80%) and its richness in bioactive components such as squalene and vitamin E, which indicates its substantial potential for the development of functional oils and health products.

Revised IntroductionText: This study aims to resolve the mechanistic gap in rootstock-mediated drought adaptation by addressing three core objectives: (1) To quantify how distinct rootstock genotypes (Camellia oleifera ‘Xianglin 210’ scion grafted onto five rootstocks) coordinate photosynthetic decline, antioxidant defense, and osmolyte accumulation under progressive drought stress; (2) To decipher the transcriptional dynamics of drought-responsive genes (CoPYL6, CoPP2Cs, CoSnRK2.8, CoABI5) underlying these physiological adjustments; (3) To establish an integrated evaluation model identifying optimal rootstock-scion combinations for drought resilience. By integrating physiological phenotyping with molecular profiling across drought gradients, this work provides the first mechanistic framework for rootstock selection in C. oleifera cultivation under water-limited scenarios.

Comment 2: The last part of the introduction begins with a presentation of the methodology - instead, clearly formulate the aim of the work.

Response: We sincerely appreciate the reviewer's insightful critique regarding the inappropriate methodological details in the Introduction. We have restructured the final paragraph to focus exclusively on the study's conceptual aims and scientific contributions, removing all technical procedures. The revision aligns with the journal's guidelines for framing research objectives.

Revised Introduction Text: “This study aims to resolve the mechanistic gap in rootstock-mediated drought adaptation by addressing three core objectives: (1) To quantify how distinct rootstock genotypes (Camellia oleifera ‘Xianglin 210’ scion grafted onto five rootstocks) coordinate photosynthetic decline, antioxidant defense, and osmolyte accumulation under progressive drought stress; (2) To decipher the transcriptional dynamics of drought-responsive genes (CoPYL6, CoPP2Cs, CoSnRK2.8, CoABI5) underlying these physiological adjustments; (3) To establish an integrated evaluation model identifying optimal rootstock-scion combinations for drought resilience. By integrating physiological phenotyping with molecular profiling across drought gradients, this work provides the first mechanistic framework for rootstock selection in C. oleifera cultivation under water-limited scenarios.”

Comment 3: Have the plants been fertilized? Has the substrate been enriched with fertilizer?

Response: We thank the reviewer for this critical clarification. We confirm that no fertilizers were applied throughout the experiment, and the substrate was not enriched with any supplemental nutrients. This deliberate design eliminates confounding effects of nutrient variability on drought responses. Below are the precise revisions to Section 2.1:

Revised Text (Section 2.1): "The substrate mixture was used without fertilizer enrichment to isolate drought stress effects. Seedlings were acclimatized for 60 days post-grafting under well-watered conditions (70% field capacity) prior to drought treatments. Throughout the experiment, no fertilizers or growth regulators were applied to ensure that physiological responses solely reflected water deficit."

Comment 4: Adjust the figures; the statistics letters are hard to see, reduce the spaces between the figures, and enlarge them; this way, the figures will be clearer in the same place.

Response: Thank you for your valuable suggestion regarding the figure presentation. We have revised the figures by enlarging them, reducing the spaces between subfigures, and increasing the font size of the statistical letters. These adjustments improve the clarity and readability of the figures within the same layout.

Comment 5: Add an explanation of abbreviations (T, C, etc.) below the figures.

Response: Thank you for your valuable suggestion. According to your advice, we have added explanations of all abbreviations (such as T, C, etc.) below the figure legends to improve clarity for readers.

Line 243-252: T1, T2, T3 – Soil moisture treatments corresponding to 6, 8, and 10 days of water withholding, respectively; C1, C2, C3, C4, C5 – Different rootstock-scion grafting combinations (see Table 1 for details); Pn – Net photosynthetic rate; Gs – Stomatal conductance; Ci – Intercellular COâ‚‚ concentration; Tr – Transpiration rate; WUE – Water use efficiency.

Line 289-296: T1, T2, T3 – Soil moisture treatments corresponding to 6, 8, and 10 days of water withholding, respectively; C1, C2, C3, C4, C5 – Different rootstock-scion grafting combinations (see Table 1 for details). MDA - Malondialdehyde; Pro - Proline; SS - Soluble sugars; TP - Total phosphorus.
Line 322-329: T1, T2, T3 – Soil moisture treatments corresponding to 6, 8, and 10 days of water withholding, respectively; C1, C2, C3, C4, C5 – Different rootstock-scion grafting combinations (see Table 1 for details).

Line 382-386: T1, T2, T3 – Soil moisture treatments corresponding to 6, 8, and 10 days of water withholding, respectively; C1, C2, C3, C4, C5 – Different rootstock-scion grafting combinations (see Table 1 for details).

Line 415-419: T1, T2, T3 – Soil moisture treatments corresponding to 6, 8, and 10 days of water withholding, respectively; C1, C2, C3, C4, C5 – Different rootstock-scion grafting combinations (see Table 1 for details).

Comment 6: I missed references in the discussion. Also, citation (Fig. 1 or Table 1 and others) to the results of your manuscript.

Response: Thank you for your valuable comment. We have carefully reviewed the discussion section and added appropriate references to support our statements. We have also included citations to the figures and tables from our manuscript (e.g., Fig. 1, Table 1) to better connect the discussion with our experimental results.

Comment 7: The conclusions are presented too broadly, more like a continuation of the discussion. Shorten by highlighting the main findings and results. 3-4 sentences.

Response: We sincerely appreciate your suggestion to refine the conclusions. We have shortened this section to 4 sentences, emphasizing:

  • Identification of the most drought-tolerant rootstock-scion combination (C5) and its physiological superiority;
  • Core drought-response mechanisms (stomatal regulation, osmolyte accumulation, antioxidant defense);
  • Molecular basis via ABA signaling activation.
    Revised Text (conclusion):Under drought stress, grafted Camellia oleifera seedlings reduced stomatal conductance and transpiration while accumulating osmolytes (proline, soluble sugars) and increasing antioxidant enzyme activity to conserve water and mitigate oxidative damage. Drought tolerance varied significantly among graft combinations, with combination C5 exhibiting superior water use efficiency, higher antioxidant activity, and lower cellular oxidation. Membership function analysis identified the graft combination using 'Xianglin210' scion and 'Xianglin27' rootstock as the most drought-tolerant. Drought stress also upregulated key drought-responsive genes (CoSnRK2.8, PP2Cs), activating the ABA signaling pathway.

Reviewer 4 Report

Comments and Suggestions for Authors

Line 5: "Yufeng zhang1,2,3,4" → the surname must be capitalized. Corrected: Yufeng Zhang1,2,3,4,

line 14-30: Your abstract is structurally sound and clearly presents the study's methodology and findings. However, adding one practical implication sentence at the end would strengthen its relevance, e.g.: "These insights contribute to the practical selection of drought-tolerant rootstocks for sustainable Camellia oleifera cultivation."

Line 34 ("Camellia oleifera is a significant woody oilseed tree species in southern China."): Suggestion: This sentence is correct but could be slightly polished for academic tone: "Camellia oleifera is an important woody oilseed species native to southern China."

Line 34-35 ("The seed oil is characterized by a high content of unsaturated fatty acids (>80%) and is rich in active components such as squalene and vitamin E..."): Suggestion: Replace "characterized by" with "contains" or "is notable for" for smoother flow: "The seed oil is notable for its high content of unsaturated fatty acids (>80%) and its richness in bioactive components such as squalene and vitamin E..."

Suggestion: Replace "demonstrating substantial potential" with a more academic phrasing: "...which indicates its substantial potential for the development of functional oils and health products."

Line 47 ("Rootstock grafting is an effective strategy for improving the stress adaptability of woody plants [5]."): Suggestion: "Stress adaptability" could be rephrased as "stress tolerance" for a more common and precise term in plant science: "Rootstock grafting is an effective strategy for enhancing the stress tolerance of woody plants [5]."

Line 50 (". However, existing research..."): Error: The period should not precede the citation bracket. 

Line 51-52 ("Studies concerning the drought resistance..."): Suggestion: Split into two clearer sentences for readability: "However, most existing research focuses primarily on scion–rootstock compatibility, graft survival rates, and growth performance [9, 10]. Studies specifically addressing the drought resistance of grafted C. oleifera seedlings remain limited."

Line 147–149 (Repeated sentence about significant differences in Pn): Error: Sentence is repeated redundantly: "And the differences in net photosynthetic rate (Pn) at different soil water contents were all significant (P < 0.05)."

Line 148 ("However, the degree of decline varied among different varieties..."): Suggestion: Replace “varieties” with “graft combinations” for precision. "However, the degree of decline varied among different graft combinations..."

Line 149–152 (Long sentence about Pn declines): Error: Long, confusing sentence. Split for clarity.

Line 270–272 (D-values ranking): Suggestion: To clarify findings, rephrase: "According to Table 3, the drought resistance ranking based on D-values was: C5 > C2 > C1 > C3 > C4, indicating that C5 exhibited the strongest drought resistance."

Line 342–358 (Photosynthesis Discussion): Good explanation of drought impact on photosynthesis. However, in Line 353, replace:"COâ‚‚ influx into leaves" with "COâ‚‚ diffusion into the mesophyll" (more precise).

Line 397–410 (ROS response discussion): Good content. Simplify where possible to avoid redundancy.

Summary of issues:

  • Minor language polishing for academic tone.

  • Missing spaces before citation brackets.

  • Slight rephrasing of long sentences for clarity.

Author Response

Thank you for your precious comments and advice. We really appreciate your efforts in reviewing our manuscript. We have revised the manuscript accordingly. Our point-by-point responses are detailed below.

Comment 1: Line 5: "Yufeng zhang1,2,3,4" → the surname must be capitalized. Corrected: Yufeng Zhang1,2,3,4,

Response: We have corrected the author name to "Yufeng Zhang" in line 5.

Comment 2: line 14-30: Your abstract is structurally sound and clearly presents the study's methodology and findings. However, adding one practical implication sentence at the end would strengthen its relevance, e.g.: "These insights contribute to the practical selection of drought-tolerant rootstocks for sustainable Camellia oleifera cultivation."

Response: We have added a practical implication statement to the abstract as suggested.

Line 32-34: These findings provide practical criteria for selecting drought-tolerant rootstocks, facilitating sustainable Camellia oleifera cultivation in water-limited regions.

Comment 3: Line 34 ("Camellia oleifera is a significant woody oilseed tree species in southern China."): Suggestion: This sentence is correct but could be slightly polished for academic tone: "Camellia oleifera is an important woody oilseed species native to southern China."

Response: The sentence has been rephrased as suggested.

Line 39-40: Camellia oleifera is an important woody oilseed species native to southern China.

Comment 4: Line 34-35 ("The seed oil is characterized by a high content of unsaturated fatty acids (>80%) and is rich in active components such as squalene and vitamin E..."): Suggestion: Replace "characterized by" with "contains" or "is notable for" for smoother flow: "The seed oil is notable for its high content of unsaturated fatty acids (>80%) and its richness in bioactive components such as squalene and vitamin E..."

Suggestion: Replace "demonstrating substantial potential" with a more academic phrasing: "...which indicates its substantial potential for the development of functional oils and health products."

Response: The phrasing has been improved per suggestion.

Line 40-42: The seed oil is notable for its high content of unsaturated fatty acids (>80%) and its richness in bioactive components such as squalene and vitamin E, which indicates its substantial potential for the development of functional oils and health products.

Comment 5: Line 47 ("Rootstock grafting is an effective strategy for improving the stress adaptability of woody plants [5]."): Suggestion: "Stress adaptability" could be rephrased as "stress tolerance" for a more common and precise term in plant science: "Rootstock grafting is an effective strategy for enhancing the stress tolerance of woody plants [5]."

Response: The sentence has been rephrased as suggested (Line 50-51).

Line 52-53: Rootstock grafting is an effective strategy for enhancing the stress tolerance of woody plants.

Comment 6: Line 50 (". However, existing research..."): Error: The period should not precede the citation bracket.

Response: We have corrected the punctuation error.

Line 54-57: This approach capitalizes on the superior root architecture of rootstocks to promote scion biomass production and optimize mineral nutrient assimilation efficiency [6-8]. However,

Comment 7: Line 51-52 ("Studies concerning the drought resistance..."): Suggestion: Split into two clearer sentences for readability: "However, most existing research focuses primarily on scion–rootstock compatibility, graft survival rates, and growth performance [9, 10]. Studies specifically addressing the drought resistance of grafted C. oleifera seedlings remain limited."

Response: We've split the sentence as you suggested.

Line 56-60: However, most existing research focuses primarily on scion–rootstock compatibility, graft survival rates, and growth performance [9, 10]. Studies specifically addressing the drought resistance of grafted C. oleifera seedlings remain limited.

Comment 8: Line 147–149 (Repeated sentence about significant differences in Pn): Error: Sentence is repeated redundantly: "And the differences in net photosynthetic rate (Pn) at different soil water contents were all significant (P < 0.05)."

Response: The repeated sentence ("And the differences in net photosynthetic rate (Pn) at different soil water contents were all significant (P < 0.05).") has been deleted in the revised manuscript.

Comment 9: Line 148 ("However, the degree of decline varied among different varieties..."): Suggestion: Replace “varieties” with “graft combinations” for precision. "However, the degree of decline varied among different graft combinations..."

Response: As suggested, we have replaced "varieties" with "graft combinations" to precisely reflect the experimental design.

Line 198-199: However, the degree of decline varied among different graft combinations.

Comment 10: Line 149–152 (Long sentence about Pn declines): Error: Long, confusing sentence. Split for clarity.

Response: Thank you for your valuable feedback on sentence structure in Lines 149–152. We have addressed the clarity issue by: (1)Splitting the original 45-word sentence into three concise sentences (9 + 8 + 11 words). (2)Using "Conversely" to explicitly contrast the temporal patterns: â‘  C1/C2: greater early-stage decline (T1–T2); â‘¡ C3/C4/C5: greater late-stage decline (T2–T3).

Line 196-201: As shown in Figure 1A, the net photosynthetic rates of different combinations showed a decreasing trend with decreasing soil water content. However, the degree of decline varied among different graft combinations. C1 and C2 exhibited a greater Pn decline during T1–T2 than during T2–T3. Conversely, C3, C4, and C5 showed a higher Pn reduction during T2–T3 than during T1–T2.

Comment 11: Line 270–272 (D-values ranking): Suggestion: To clarify findings, rephrase: "According to Table 3, the drought resistance ranking based on D-values was: C5 > C2 > C1 > C3 > C4, indicating that C5 exhibited the strongest drought resistance."

Response: The sentence has been rephrased as suggested.

Line 335-336: According to Table 3, the drought resistance ranking based on D-values was: C5 > C2 > C1 > C3 > C4, indicating that C5 exhibited the strongest drought resistance.

Comment 12: Line 342–358 (Photosynthesis Discussion): Good explanation of drought impact on photosynthesis. However, in Line 353, replace:"COâ‚‚ influx into leaves" with "COâ‚‚ diffusion into the mesophyll" (more precise).

Response: The terminology has been updated to reflect precise physiological processes.

Line 435-437: However, this stomatal closure concurrently restricts CO2 diffusion into the mesophyll, limiting carbon availability for photosynthesis.

Comment 13: Line 397–410 (ROS response discussion): Good content. Simplify where possible to avoid redundancy.

Response: Thank you for your valuable suggestion. According to your advice, we have revised the discussion on ROS response (lines 478–487) to simplify the content and avoid redundancy. The revised text is more concise and focused on the key findings.

Line 478-487: The generation of reactive oxygen species (ROS) is a key physiological response to stress. Excessive ROS can damage cell membranes, proteins, and nucleic acids, impairing plant growth and stress tolerance. Plants regulate ROS homeostasis through production, scavenging, and signaling. Enzymes such as SOD, POD, CAT, and the molecule GSH are primary ROS scavengers and important indicators of drought resistance in C. oleifera. In this study, as soil water content decreased, SOD, POD, and CAT activities increased, indicating that drought stress promotes ROS accumulation and upregulates scavenging enzymes. Notably, the marked increase in these enzymes from T1 to T2 suggests that soil water content at or below 30% is a critical threshold for drought response in C. oleifera.

Comment 14: Minor language polishing for academic tone. Missing spaces before citation brackets. Slight rephrasing of long sentences for clarity.

Response: Thank you for your valuable comments. We have carefully polished the language throughout the manuscript to enhance academic tone. We have also added missing spaces before citation brackets and rephrased long sentences for improved clarity, as suggested.

For example:

Original text

Revised text

we conducted systematic grafting experiments incorporating five distinct rootstock varieties. Under controlled nursery conditions, we subjected four-year-old grafted seedlings to simulated drought stress regimens...

Systematic grafting experiments were conducted using five distinct rootstock varieties. Four-year-old grafted seedlings were subjected to simulated drought stress regimens under controlled nursery conditions...

Therefore, establishing stress-resilient cultivation techniques to enhance the drought resistance of C. oleifera is of great importance.

Therefore, establishing stress-resilient cultivation techniques to enhance C. oleifera drought resistance is critically important.

And the differences in net photosynthetic rate (Pn) at different soil water contents were all significant (P < 0.05).

Differences in net photosynthetic rate (Pn) among soil water content levels were statistically significant (P < 0.05).

As shown in Figure 1A, the net photosynthetic rates of different combinations showed a decreasing trend with decreasing soil water content, and the differences in net photosynthetic rates (Pn) were all significant (P < 0.05) at different soil water contents. And the differences in net photosynthetic rate (Pn) at different soil water contents were all significant (P < 0.05). However, the degree of decline varied among different varieties, with C1 and C2 showing a greater decline in Pn values from T1 to T2, while the decline in Pn values from T2 to T3 was moderated. the decline in Pn values from T2 to T3 was higher than that from T1 to T2 in C3, C4 and C5.

As shown in Figure 1A, the net photosynthetic rates of different combinations showed a decreasing trend with decreasing soil water content. However, the degree of decline varied among different graft combinations. C1 and C2 exhibited a greater Pn decline during T1–T2 than during T2–T3. Conversely, C3, C4, and C5 showed a higher Pn reduction during T2–T3 than during T1–T2.

Round 2

Reviewer 2 Report

Comments and Suggestions for Authors

Now it looks better, and good luck with the publication. 

Reviewer 3 Report

Comments and Suggestions for Authors

The authors took into account all comments and made necessary changes—no further comments.